# Performance Validation and Calibration Conditions for Novel Dynamic Baseline Tracking Air Sensors in Long-term Field Monitoring

Han Mei[1], Peng Wei[2*], Meisam Ahmadi Ghadikolaei[1], Nirmal Kumar Gali[1], Ya Wang[1], Zhi Ning[1,*]

[1]Division of Environment and Sustainability, The Hong Kong University of Science and Technology, Hong Kong, China
[2]College of Geography and Environment, Shandong Normal University, Jinan, China

*Correspondence to*: Zhi Ning (zhining@ust.hk), Peng Wei (pengwei@sdnu.edu.cn)

**Abstract.** The rapid expansion of low-cost sensor networks for air quality monitoring necessitates rigorous calibration to ensure data accuracy. Despite numerous published field calibration studies, a universal and comprehensive assessment of factors affecting sensor calibration remains elusive, leading to potential discrepancies in data quality across different networks. This study deployed eight sensor-based monitors in strategically chosen locations continuously for two years in Hong Kong, Macau, and Shanghai. These locations covered a wide range of climatic conditions: Hong Kong's subtropical climate, Macau's similar yet distinct urban environment, and Shanghai's more variable climate. Each monitor employed a dynamic baseline tracking method for the gas sensors, which isolates the concentration signals from temperature and humidity effects, enhancing the sensors' accuracy and reliability. This strategic deployment ensured that the sensors' performance and calibration processes were tested across diverse atmospheric conditions. The tests, which involved evaluating the validation performance by analyzing randomly selected calibration sample subsets ranging from 1 to 15 days, indicated that the length of the calibration period, pollutant concentration range, and time averaging period are pivotal for sensor calibration quality. We determined that a 5–7 days calibration period minimizes calibration coefficient errors, and a wider concentration range improves the validation $R^2$ values for all sensors, suggesting the necessity of setting specific concentration range thresholds. A time averaging period of at least 5 minutes for data with 1-minute resolution was recommended to enable optimal calibration in field operation. This study emphasizes the need for a comprehensive calibration assessment and the importance of considering environmental variability in sensor calibration condition. These findings offer methodological guidance for the calibration of other sensor types, providing a reference for future research in the field of sensor calibration.

## 1 Introduction

Rapid advancements in low-cost air sensor technology have led to a significant increase in their applications across various fields. These sensors offer a promising and cost-effective solution for monitoring air pollution at finer spatial scales and in novel locations compared to traditional monitoring methodologies. This has resulted in a growing demand for high-quality sensor data. Calibration is an indispensable component of the air sensor operational paradigm, pivotal for securing accurate

and dependable data. By establishing a relationship between the raw sensor output and the corresponding reference measurement, calibration enhances the accuracy and precision of sensor data.

Common calibration methods include multi-point calibration with standard gases, controlled chamber calibration (Sousan et al. 2016; Papapostolou et al. 2017), on-site probe gas calibrations, and field side-by-side calibration (Bisignano et al. 2022; Holstius et al. 2014; Spinelle et al. 2015; 2017). The first three methods are laboratory-based methods or rely on standard gas, which inherently possess constraints and may not fully capture the intricate interactions of multiple pollutants and environmental factors encountered in situ. This limitation raises concerns about the applicability of calibration results obtained under controlled conditions to actual monitoring environments(Castell et al. 2017). An alternative approach is the side-by-side calibration, which involves the co-locating sensor systems with reference analyzers in real-world environmental settings for a designated duration. This approach leverages the natural fluctuation of pollutant concentrations and environmental factors to accurately calibrate the sensors' sensitivity and baseline response. It is advantageous due to its procedural simplicity, negligible consumable usage, and cost efficiency compared to laboratory assessments (Castell et al. 2017). Consequently, it has become as a preferred method for calibration in various scenarios (Spinelle et al. 2015; 2017).

Despite the widespread application of field side-by-side calibration, several critical concerns persist regarding the process. The primary issue is the selection of appropriate calibration conditions. Factors like the calibration duration (Levy Zamora et al. 2023), the pollutant concentrations distribution (Levy Zamora et al. 2023), sensor ageing(Li et al. 2021), interference from non-target gases (Cross et al. 2017), the impacts of temperature and relative humidity (Ariyaratne et al. 2023), and various gas sampling methods can significantly influence the calibration results. Determining the optimal conditions is crucial for achieving accurate and reliable calibration results. Extensive research has focused on the calibration period, the most frequently reported in recent studies (Datta et al. 2020; Gao, Cao, and Seto 2015; Kim et al. 2018; Mukherjee et al. 2019; Pinto et al. 2014; Spinelle et al. 2015; 2017; Topalovic et al. 2019). One study by Zamora et al. (2023) evaluated the impact of calibration period on calibration quality using calibration periods of up to 6 months from one year of $PM_{2.5}$, CO, NO, $NO_2$, and $O_3$ data in Maryland, US. Their results indicated diminishing improvements in median root-mean-square error (RMSE) for calibration periods longer than six weeks for all sensors. Zamora et al. (2023) also highlighted the importance of considering environmental conditions during the calibration period that are similar to those encountered during the evaluation period to achieve the best calibration performance. Another study by Okorn et al. (2021) reported that longer calibration periods (i.e., six weeks) resulted in fits with a reduced bias compared to fits obtained from shorter calibration periods (1 week), while the one-week calibrations yielded the best $R^2$ (coefficient of determination) values. While these studies have offered valuable insights into sensor field calibration conditions, more discussion is needed on other calibration factors, particularly the range of pollutant concentrations during the calibration period and the selection of time averaging length for raw data before calibration. These two factors are more straightforward to standardize and quantify compared to other factors, as they can be defined with specific numerical values and consistent measurement protocols, making it easier to compare results across different studies and ensure reliable calibration outcomes.

In addition to investigating calibration conditions, an equally crucial aspect to address is the development of an effective calibration model that can accommodate these optimized sensor calibration conditions. This study focuses on electrochemical sensors, which are the most common type of air quality gas sensors. Laboratory studies of commercial electrochemical sensors have shown linear correlations between current response and gas analyte concentration under stable temperature and relative humidity (RH) conditions (Mead et al. 2013; Collier-Oxandale et al. 2020; Wei et al. 2018; Zong et al. 2021). However, due to their electrochemical characteristics, these sensors often exhibit non-linear responses to variations in temperature and RH (Wei et al. 2018; Ariyaratne et al. 2023; Li et al. 2021), which can significantly impair their performance in real-world applications. In the past, most studies have adopted generic multiple linear regression (MLR) or machine learning models to calibrate raw sensor data, taking into account various complex variables such as temperature, RH, their gradient and cross-sensitivity to other pollutants (Datta et al., 2020; Han et al., 2021; Levy Zamora et al., 2023; Si et al., 2020; Topalovic et al., 2019; Wei et al., 2020; Zimmerman et al., 2018). These models, while comprehensive, often face limitations such as the risk of over-fitting, extensive training requirements, restricted applicability, and difficulties in replicating and scaling up for large sensor numbers. Furthermore, the complexity of machine learning models can pose significant barriers for everyday users.

Instead of relying solely on mathematical algorithms for sensor calibration, we assessed a novel dynamic baseline tracking technology designed to physically mitigate temperature and RH effects on sensor signals, allowing these kind of gas sensor devices, i.e. Mini Air Stations (MAS, Sapiens), to output sensing data most directly related to the concentration signal. By isolating the non-linear influences of temperature and RH on sensor readings, this technology allowed us to focus exclusively on optimum calibration strategy and enabled the development of a refined linear calibration model. Based on the linear calibration model, we further identify the critical factors that influence calibration quality, thus optimizing calibration conditions for $NO_2$, NO, CO, and $O_3$ electrochemical sensors. Our research uncovers three pivotal factors that significantly impact sensor calibration and validation performance: calibration period, concentration range, and time averaging. By examining these factors' effects on the variation of sensor's calibration coefficients, we aim to deepen the understanding of sensor calibration processes and enhance the performance of low-cost electrochemical air sensors. This methodology not only simplifies the calibration process but also ensures that the calibration model remains robust and applicable in varied and long-term field conditions.

## 2 Material and methods

### 2.1 Data collection

#### 2.1.1 Sensor devices

Eight microsensor-based Mini Air Stations (MAS-AF300, Sapiens), hereinafter referred to as 'MAS', shown in Figure 1, were utilized in this study for continuous measurements of the air pollutants $NO_2$, NO, $O_3$, and CO under field conditions. Each MAS unit included three or four gas sensors along with a combined RH and temperature sensor (SHT-75, Sensirion AG). This

study focuses on electrochemical gas sensors for NO$_2$ (Alphasense NO$_2$-B43F), NO (Alphasense NO-B4), CO (Alphasense CO-B4), and O$_3$ (Alphasense OX-B431). Please note that the O$_3$ concentration is determined by calculating the difference between the readings of the oxidizing gas sensor (OX-B431) and the NO$_2$ sensor (NO$_2$-B43F). Furthermore, the MAS system incorporates numerous sophisticated functionalities. All the gas sensors are equipped with the dynamic baseline tracking technology by the manufacturer with details in the following section. The system is also equipped with an active air sampler, ensuring a flow rate of 0.8 L min$^{-1}$. The sample air undergoes filtration through a Teflon dust filter before directly entering the sensor module, without the implementation of any temperature or humidity control measures. The Teflon dust filter for each MAS will be replaced regularly every month to prevent dust from entering the gas module and causing measurement errors and shortening the sensor life. To mitigate potential drift during long-term deployment, the MAS gas module incorporates an auto-zeroing function. During the zeroing process, the gaseous pollutant measurement module receives air samples from a separate zero module, from which NO, NO$_2$, and O$_3$ have been significantly mitigated. The data collected during the zeroing period is subsequently analyzed to rectify any drift effects during the long-term deployment phase, as part of the data cleaning procedure. A comprehensive description of this technology and its functional advantages can be found in a paper by Sun (Sun, Westerdahl, and Ning 2017). All these incorporated functionalities in the MAS system are aimed at optimizing sensor performance, enhancing measurement accuracy, and ensuring their long-term stability.

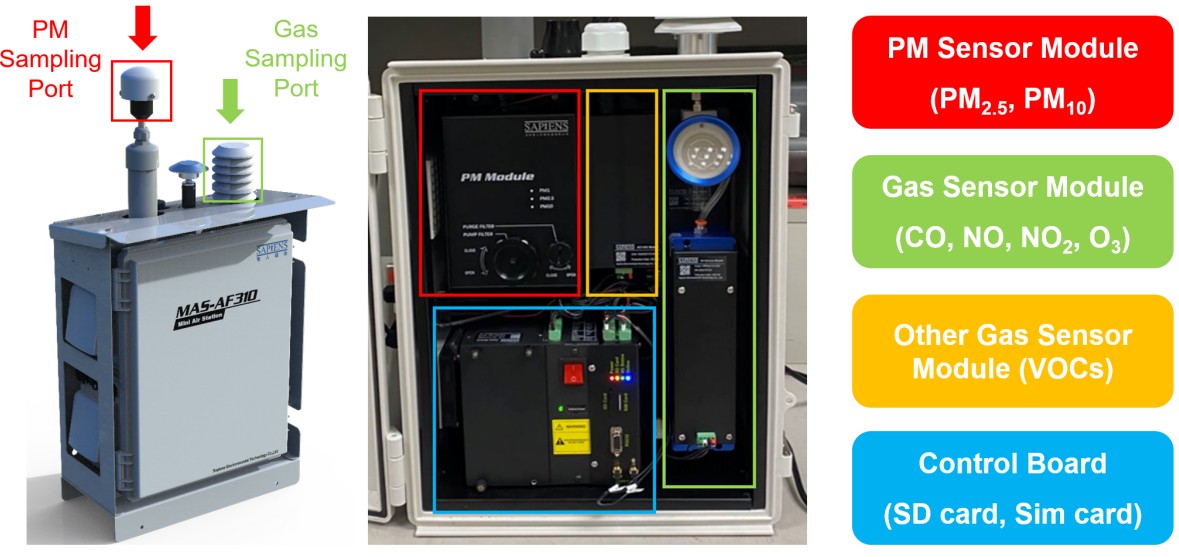

**Figure 1. Structure diagram of MAS monitoring devices (dimensions: 420 × 320 × 180 mm, H × W × D; weight: 12 kg; power consumption: 15W).**

### 2.1.2 Measurement campaign details

To assess sensor performance under varying ambient conditions, these MASs were deployed in three distinctively different urban and climatic settings: Hong Kong's humid subtropical climate, Macau's somewhat similar yet distinct urban environment, and Shanghai's more variable climatic conditions. Each city featured a co-location campaign with an AQMS, as detailed in Table 1, and the AQMSs were equipped with Federal Equivalent Method (FEM) reference analyzers.

The first co-location campaign in Hong Kong involved the four MASs, each equipped with all four types of gas sensors ($NO_2$, NO, CO, and $O_3$), which were placed at the Tseung Kwan O AQMS (22.3716°E,114.1148°N) regulated by the Hong Kong Environmental Protection Department. This station serves as a representative urban site, providing conditions suitable for sensor evaluation in a complex urban environment. In the second co-location campaign, two MASs were located at the Taipa Air Quality Monitoring Station (22.15896°E, 113.56882°N) in Macau, focusing on $NO_2$, NO, and $O_3$ to capture the general urban background conditions unique to the region. The third campaign took place in Shanghai, where two MASs, monitoring $NO_2$, NO, and CO, were placed separately alongside two sets of reference analyzers at the Waigaoqiao Port 2 site (31.36662°E, 121.57242°N) and Port 4 site (31.33302°E, 121.65496°N). This campaign was also the longest co-location campaign, lasting 22 months, offering a prolonged observation of the diverse and more polluted air quality conditions typical of a major industrial hub. These locations were chosen to ensure a comprehensive analysis across a spectrum of urban pollution levels and environmental conditions.

All eight MAS units were designed to automatically transmit the measured raw sensor signals and concentration data of the pollutants from the MAS to a secure cloud server in real-time at 1-minute resolution. The reference analyzer in Hong Kong provided 1-minute time resolution pollutant concentration data, while those in Macau and Shanghai provided hourly averaged data, enabling us to conduct calibration analysis at varying time resolutions.

**Table 1. Details of MAS devices in co-location calibrations.**

| Location | MAS ID | Reference analyzer data time resolution | Co-location periods | Monitoring pollutants and concentration range (5th to 95th percentile range) | MAS inside temperature and RH range |
|---|---|---|---|---|---|
| Hong Kong | MAS1 | Minute | 2021-07-27 00:00 to 2022-10-10 00:00 (15 months) | $NO_2$: 3.7 ppb - 34.6 ppb NO: 0.4 ppb - 18.0 ppb CO: 152 ppb - 643 ppb $O_3$: 4.3 ppb - 69.1 ppb | Temp: 10 ˚C - 43˚C RH: 17% - 85 % |
|  | MAS2 | Minute | 2021-12-24 00:00 to 2022-10-10 00:00 (10 months) |  | Temp: 10 ˚C - 46˚C RH: 16% - 86 % |

| | MAS3, MAS4 | Minute | 2021-07-10 00:00 to 2022-10-10 00:00 (15 months) | | Temp: 10 ˚C - 45˚C RH: 16% - 93 % |
|---|---|---|---|---|---|
| Macau | MAS5, MAS6 | Hourly | 2021-04-04 13:00 to 2022-04-26 05:00 (13 months) | $NO_2$: 0 ppb - 26.3 ppb NO: 0 ppb - 17.6 ppb $O_3$: 0 ppb - 68.8 ppb | Temp: 10 ˚C - 47˚C RH: 21% - 89 % |
| Shanghai | MAS7, MAS8 | Hourly | 2019-10-12 01:00 to 2021-07-31 23:00 (22 months) | $NO_2$: 14.1 ppb - 63.4 ppb NO: 3.2 ppb - 142.5 ppb CO: 258 ppb - 862 ppb | Temp: -8 ˚C - 51˚C RH: 0% - 90 % |


## 2.2 Dynamic baseline tracking method to mitigate environmental effects on sensors

The sensor device (MAS, Sapiens) has deployed a novel gas sensing technology that enables the isolation of the concentration signal from environmental variables of temperature and RH through a patented dynamic baseline tracking method by the manufacturer, which operates by differentiating the varying environment and target pollutant induced sensor signals using a
dual-sensor module. Figure 2 shows the conceptual diagram of MAS sensor module and general working principle of the dynamic baseline tracking method. This gas sensor system comprises a primary sensor – that is directly exposed to the air, capturing the original signal (designated as ORG) influenced by varying pollutants, temperature, and RH - and a proprietary pair differential filter sensor (designated as PDF) to track the dynamic baseline signal driven only by temperature and RH. The PDF sensor is equipped with a water molecule permeable membrane that allows the water vapor to penetrate through while
filtering out the target gas modules from entering the sensor head. The differential signal (measured in volts) between the ORG and PDF sensors decouples the temperature and humidity effects, yielding a pure signal that reflects target gas concentrations. Each MAS sensor module produces four distinct outputs for a specific pollutant: (i) the ORG sensor signal in volts, $V_{ORG}$, (ii) the PDF sensor signal in volts, $V_{PDF}$, (iii) the voltage output from the difference of the ORG and PDF sensor signals in volts, $V_{DIFF}$, and (iv) the concentration output of target gas in ppb, $Conc$. Each MAS has an onboard algorithm capability that converts
sensor signals to concentration, with the conversion automatically performed onboard the MAS for real-time concentration output. Eq. (1) presents the conversion equation for $NO_2$, NO, and CO, where '$a$' denotes the slope of the equation, which is also indicative of the sensitivity (ppb mV$^{-1}$) of the electrochemical sensors, and '$b$' represents the intercept of the equation. For the gas sensors exhibit cross-sensitivity with non-target gases, an interfering gas correction component can be incorporated. Eq. (2) presents the equation for calculating $O_3$ concentrations using the Alphasense OX-B431 sensor with $NO_2$ as an
interferent. The coefficient '$f$' accounts for the cross-interference from $NO_2$, and our empirical data, derived from a substantial number of tests, indicates that '$f$' typically falls within the range of 0.8 to 1.2.

$$Conc(NO_2, NO, CO) = a \times V_{DIFF} + b \,, \tag{1}$$

$$Conc(O_3) = a \times V_{DIFF} + b - f \times Conc(NO_2), \tag{2}$$

Prior to initiating the co-location campaign, a 15-day pre-test under field conditions and a laboratory test in the environmental
chamber were conducted to demonstrate the method's capability to enhance the sensor performance under varying temperature
and humidity conditions.

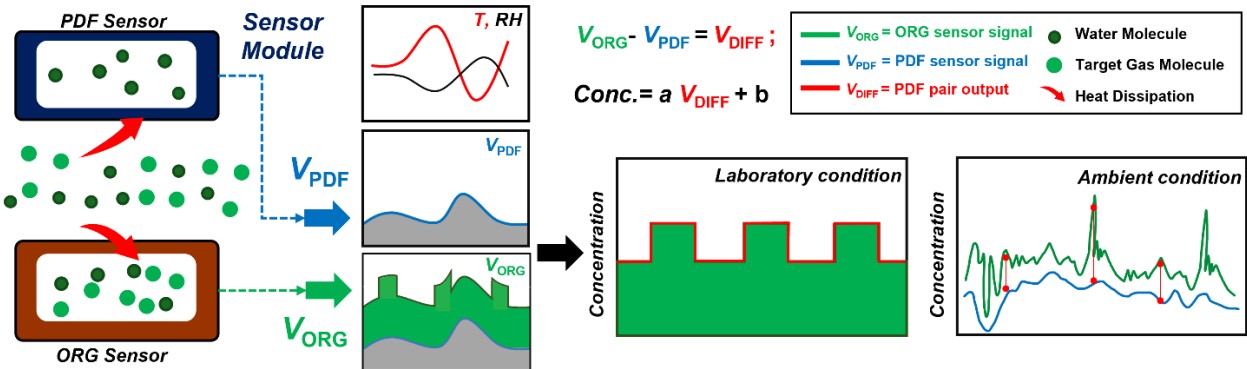

**Figure 2. A conceptual diagram of the PDF-enabled MAS sensor device. In laboratory tests, standard gas with constant
concentrations is periodically injected into the PDF and ORG sensors throughout varying temperature and RH cycles to investigate**
**their effects on the sensor performance. The PDF tracks the baseline signal driven only by temperature and RH, while the ORG
sensor captures the concentration profile influenced by both the target gas module and environmental conditions. The differential
signal between the ORG and PDF sensors decouples the baseline signal induced by temperature and RH, producing a pure signal
that reflects the target gas concentrations. This concept is also applicable to ambient conditions, where the differential signal between
the paired ORG and PDF sensors demonstrates the accuracy and robustness of PDF technology for ambient air monitoring.**


### 2.3 Impact analysis of three crucial factors on calibration conditions

This study specifically focuses on conducting field tests to identify optimal calibration conditions by examining three primary
factors that influence sensor calibration performance: (a) calibration period duration; (b) concentration variation range; and (c)
time averaging pre-processing.

**Calibration Period Optimization**

Calibration is typically conducted within a specific timeframe, constrained by time and resource availability. Standard
protocols involve calibrating sensors over durations ranging from a few days to several weeks prior to their utilization in field
monitoring applications. The calibration's effectiveness largely depends on this timeframe, referred to as the calibration period.
The calibration period test in this study uses subsets of the full co-location period to generate a range of hypothetical calibration
periods. We investigated calibration period scenarios ranging from 1 to 15 days. In each scenario, 500 samples were randomly

selected using the numpy.random.choice() function in Python, ensuring randomness and independence in the selection of co-location timing. This approach is intended to create hypothetical scenarios that reflect the diverse conditions and variability under which calibration might occur in real-world sensor calibration practices. Sample sizes of 250, 500, and 1000 were tested, results stabilized with 500 samples, indicating minimal impact from decreasing or increasing the sample size further. The 500 randomly selected calibration periods were illustrated in Figure S1 in the supplementary materials, which shows the start times for these periods for NO, with the approach also applied to $NO_2$, CO, and $O_3$ sensors.

These calibration samples were used as the training set for each hypothetical calibration period in the calibration model to evaluate the range of potential $R^2$ and RMSE when applied in the sensor validation periods. Firstly, these samples were standardized to hourly data to facilitate consistent comparisons across various MAS units. The calibration coefficients (slope and intercept) of these samples were calculated as per Eq. (1) or Eq. (2). Subsequently, these coefficients were validated using the following month's data by comparing the hourly calibrated sensor data and hourly reference data. A superior validation performance, indicated by higher $R^2$ values and lower RMSE values, suggests that the calibration period effectively captures the relationship between the calibrated sensor and the reference data, thereby indicating an optimal calibration duration. This evaluation was not limited to the calibration period's immediate outcome; it also included a comparison of $R^2$ and RMSE metrics against the hourly data validation set from the subsequent month. This dual-phase evaluation underscores that the calibration's true merit is better judged during the post-calibration validation phase, adhering to the standard practice of a bounded calibration period followed by an extended validation phase.

**Concentration Range Analysis**

We propose the hypothesis that users can strategically select a co-location period to minimize the calibration duration, recognizing that the calibration period is not the sole factor to consider when optimizing instrument co-location for calibration purposes. A critical aspect is to evaluate the representativeness of environmental conditions during the calibration period in relation to those observed during the long-term evaluation periods. Since the influence of temperature and RH on sensor signals has been significantly mitigated, concentration emerges as the key factor that accurately reflects environmental conditions. To analyze how the range of pollutant concentrations during the calibration period affected the sensor validation performance, we compared the validation $R^2$ and RMSE outcomes with the same calibration period length but varied concentration ranges.

Firstly, we segmented the samples into distinct categories based on their concentration ranges while maintaining a constant calibration period. We employed the $5^{th}$ to $95^{th}$ percentile of the pollutant concentration in each category to define each range. This approach mitigates the impact of sporadic peak values, ensuring they do not disproportionately affect the overall concentration range assessment. Subsequently, the effectiveness of calibration across these ranges was systematically evaluated by comparing $R^2$ and RMSE metrics during the validation periods in the subsequent month. This strategy enabled a thorough examination of how the concentration range impacts calibration accuracy, providing insights into the optimal range needed for precise sensor calibration.

**Time Averaging Evaluation**

We also evaluated the influence of time averaging on calibration efficacy to identify the optimal data resolution for the best calibration outcomes. Given that reference analyzers and sensors can provide data at granular levels, down to minutes or seconds, pre-calibration data processing plays a crucial role in the accuracy of calibration.

In this time averaging analysis, we compared the calibration performance of data averaged over different time intervals, from minutes to hours. After processing the calibration data set with varied time averaging intervals, the resulting calibration coefficients were evaluated against the data from the following month's validation set. For example, for a sample with calibration period of 1 day, sensor and reference data were averaged over 1/3/5/7/9/11/30/60/120/180 minutes and used to determine the sensor coefficients for each time averaging interval. Following that, these coefficients were independently applied to the following one-month validation period with hourly data, to determine the $R^2$ and RMSE under each time averaging intervals. The ideal time averaging interval was determined based on the highest $R^2$ and lowest RMSE values obtained in this validation phase, pinpointing the most effective time resolution for calibration.

## 3 Results and discussion

### 3.1 MAS sensor performance against temperature and RH variability

Before initiating the long-term co-location campaign, the MAS units equipped with $NO_2$, $NO$, $CO$, and $O_3$ sensors were tested in Hong Kong, demonstrating the dynamic baseline tracking method's ability to enhance electrochemical sensor performance against varying temperatures and RH. We tested four MAS units and presented findings from this one MAS as an example to evaluate the robustness of the PDF technology. During the 15-day pre-test in the summer (June 1-15, 2019), temperatures varied between 28 ˚C and 42 ˚C, with RH levels from 45% to 87%. The outputs from the PDF sensor, the ORG sensor, and the differential output between the paired ORG - PDF sensor are illustrated separately in Figure 3(a)-(d). The voltage signals from the PDF and ORG sensors were converted into concentration outputs using coefficients derived from Eq. (1). As shown in the figure, even during the typical ambient concentration ranges, the accuracy of the ORG sensor outputs for gases other than CO was notably poor, primarily due to significant influences from field temperature and RH. It was observed that the PDF sensor outputs for all gas pollutants did not exhibit a linear relationship with temperature or RH profiles. Different sensor types demonstrated distinct response patterns to variations in temperature and RH, highlighting the complex non-linear characteristics of electrochemical sensors in relation to baseline dependence on these environmental factors.

With the PDF enabled sensors, the physical separation of the climatic driven baseline and target gas driven sensitivity is demonstrated to be feasible and effective. By subtracting the output of the PDF sensor from that of the ORG sensor, the resulting ORG – PDF output reveals a clear gas concentration profile that aligns closely with reference measurements. This relationship is illustrated in the scatter plots presented in Figure 3(f)-(i). For $NO_2$, the ORG – PDF sensors showed stronger performance, with a high $R^2$ (0.99) and low RMSE (0.94), compared to the lower $R^2$ (0.44) and higher RMSE (5.80) for the ORG sensors without the PDF module. For NO and $O_3$, the ORG – PDF sensors also demonstrated stronger performance compared to the ORG sensors without the PDF module. Specifically, the ORG – PDF sensors had strong $R^2$ (0.97 for both NO

and $O_3$) and low RMSEs (1.72 for NO, 1.05 for $O_3$), while the ORG sensors without the PDF module had weaker $R^2$ (0.73 for NO, 0.59 for $O_3$) and higher RMSEs (5.37 for NO, 4.18 for $O_3$). For CO, the sensors exhibited comparable performance, with $R^2$ around 0.93-0.94 and RMSE values between 16.70-19.00, regardless of the PDF module. We tested four MASs and the other PDF enabled sensors were shown in Figure S2. Their data quality performance has been consistent with the findings reported data here. These significant discrepancies between the ORG sensor output and ORG – PDF sensor output, especially for NO, $NO_2$, and $O_3$, highlight the importance of the dynamic baseline tracking method in improving the accuracy and reliability of measurements, notably under low concentration conditions influenced by temperature and RH.

Additionally, laboratory tests in environmental chambers assessed the MAS NO sensor (Figure S3), exposing it to broad temperature (0°C to 30°C) and RH (10% to 90%) ranges. Despite these fluctuations, MAS sensors maintained consistent and stable readings after applying the dynamic baseline tracking method, as shown in Figure S3(b), with concentration steps from 50 to 300 ppb. The outcomes from both field and laboratory tests confirm that the dynamic baseline tracking method effectively neutralizes temperature and RH effects, primarily for $NO_2$, NO, and $O_3$ sensors, achieving desired performance while focusing primarily on concentration factors for subsequent analysis. Similar pre-tests were also conducted with the MAS units in Macau and Shanghai to assess the effectiveness of the dynamic baseline tracking method.

Upon completion of the pre-tests, the long-term field co-location campaigns were initiated. The dynamic baseline tracking method was first evaluated in this study to prove its effectiveness in long-term field tests. The performance of MAS1, particularly for NO and $NO_2$, throughout the campaign, was depicted in Figures S4 and S5. It should be noted that a single fixed calibration coefficient was used throughout the entire campaign duration. This fixed coefficient enabled the calibrated sensor data to consistently perform well throughout the co-location campaign. The absolute error (sensor - reference) generally stayed within ± 5ppb, and the relative error (absolute error/reference) was primarily under 15%, indicating effective mitigation of temperature and RH impacts on the sensor's output, even during extended field conditions over a year. Importantly, the long-term analysis in Figures S4 and S5 showed that selecting suitable calibration coefficients can ensure the sensors' stability and accuracy over prolonged periods. However, dedicating several months or even up to a year for calibration is not feasible in standard practice. Therefore, our main goal is to determine the optimal coefficients from short-term calibration periods to enhance long-term validation performance.

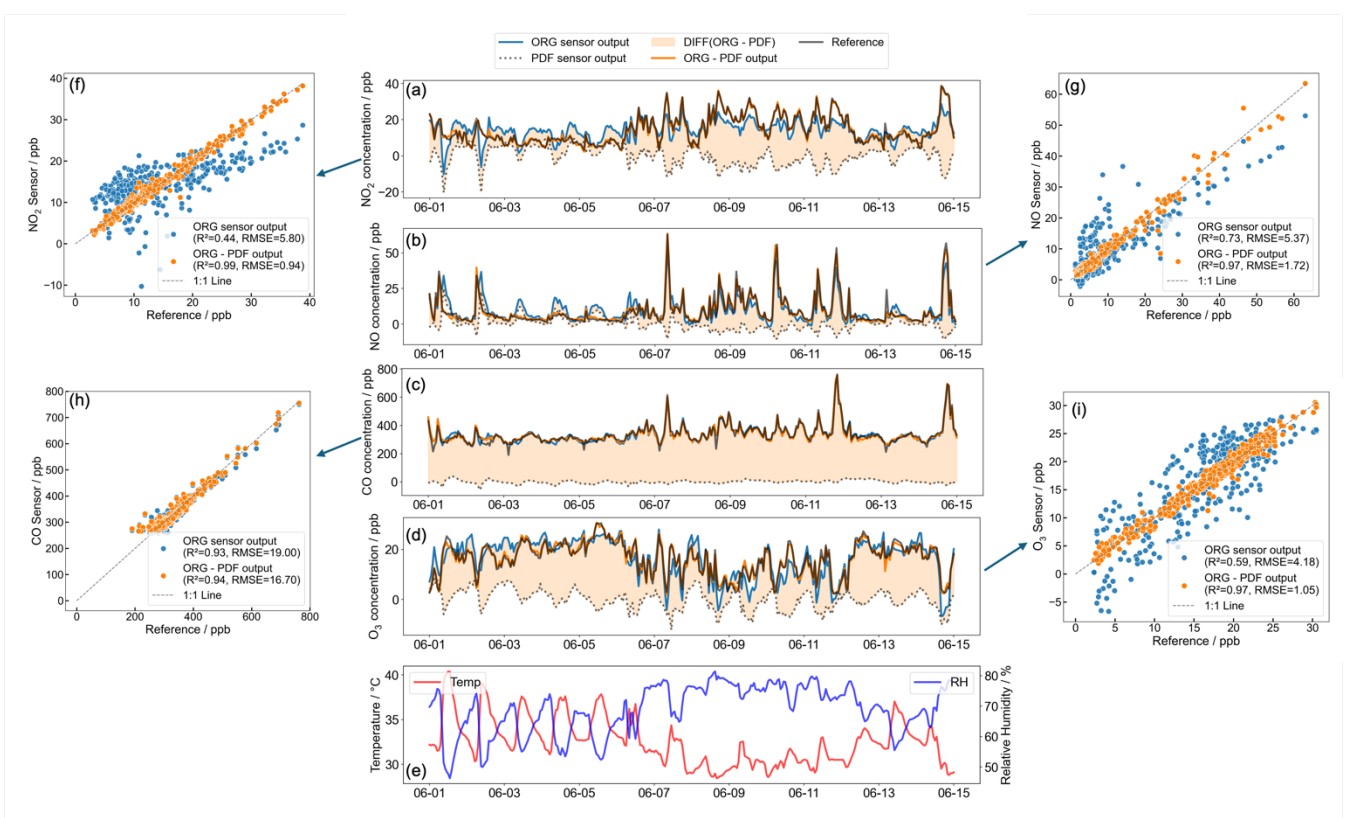

**Figure 3. (a-d) Performance validation of the MAS's ORG and PDF sensors for detecting NO₂, NO, CO, and O₃ under field conditions in 2019. (e) Displays the temperature and RH measured inside the MAS gas sensor modules. (f-i) Compares the readings from the ORG sensor and the MAS PDF-enabled sensor with reference measurements.**

## 3.2 Impact of calibration period on sensor calibration

As detailed in Section 2.3, we used 500 randomly selected samples for each calibration period, and this process generated 500 sets of calibration slopes and $R^2$ / RMSE values from the validation period. Figure S6 displays the median and the 25$^{th}$ to 75$^{th}$ percentile range of these $R^2$ / RMSE results across all eight MAS units with NO₂ and NO sensors and all six units with CO and O₃ sensors. Figure 4 extracts the 25$^{th}$ to 75$^{th}$ percentile of each MASs results and combines them into a boxplot, making the trend across the calibration period more apparent. An increase in the median of $R^2$ (e.g. for NO, $R^2$ improved from 0.83 to 0.95 as the calibration period went from 1 to 15 days) coupled with a reduction in the median of RMSE (e.g. for NO, RMSE decreased from 3.71 to 2.12 over the same calibration period) shown in Figure 4 indicate improved validation performance. The narrowing of the 25$^{th}$ to 75$^{th}$ percentile range across calibration periods (e.g. for NO, $R^2$ range tightened from 0.66-0.96 to 0.90-0.98 as the calibration period went from 1 to 15 days) further supports this, with a tightening of validation performance towards a steadier state and reduced chance of abnormal calibration.

In Figure 4, the most notable enhancements in validation performance were observed within the initial 1 to 3 days. Beyond this period, the rate of improvement was found to be less clear, with the median $R^2$ increasing by less than 0.02 and the median RMSE decreasing by less than 0.1 (but less than 1 for CO) for further increases in the calibration period. For $NO_2$, NO, and $O_3$, the upward trend in validation $R^2$ and the downward trend in RMSE were observed, plateauing after 5 days. CO sensors in most MAS units reach stable $R^2$ after 7 days. This suggests lengthening the calibration period beyond 5 days for $NO_2$, NO, $O_3$ or 7 days for CO does not markedly benefit sensor data performance. If the sensor users can strategically select the co-location period to minimize the calibration duration, a period of 5–7 days is identified as most effective for minimizing errors in calibration coefficient and avoiding notably low validation $R^2$ values.

The aforementioned results are based on an average pattern derived from the combined data of all sensors. Figure S6 presents the separate performance of all eight MAS sensors over varying calibration periods. The $NO_2$, NO, CO, and $O_3$ sensors in MAS1-4 in Hong Kong and MAS5-6 in Macau exhibited trends consistent with those shown in Figure 4. A noteworthy observation in Figure S6(a)-(b) is that the $NO_2$ and NO sensors in MAS7 and MAS8 of Shanghai campaign showed consistent performance over all calibration periods, lacking the trends observed in Figure 4. Considering that the NO and $NO_2$ concentrations in the site of Shanghai are significantly higher than those in the other two cities, it is hypothesized that the elevated pollutant concentrations at the Shanghai port provided a more favorable calibration condition, thereby diminishing the contribution of the calibration period. Thus, we conclude that for calibration condition with a narrower concentration range, a calibration period of at least 5 to 7 days is necessary, whereas more polluted ambient environments are more conducive to sensor concentration calibration. Despite the short calibration duration of 1–3 days, the extensive concentration range assessed contributed to more precise calibration coefficients and improved validation performance, as will be discussed in the next section.

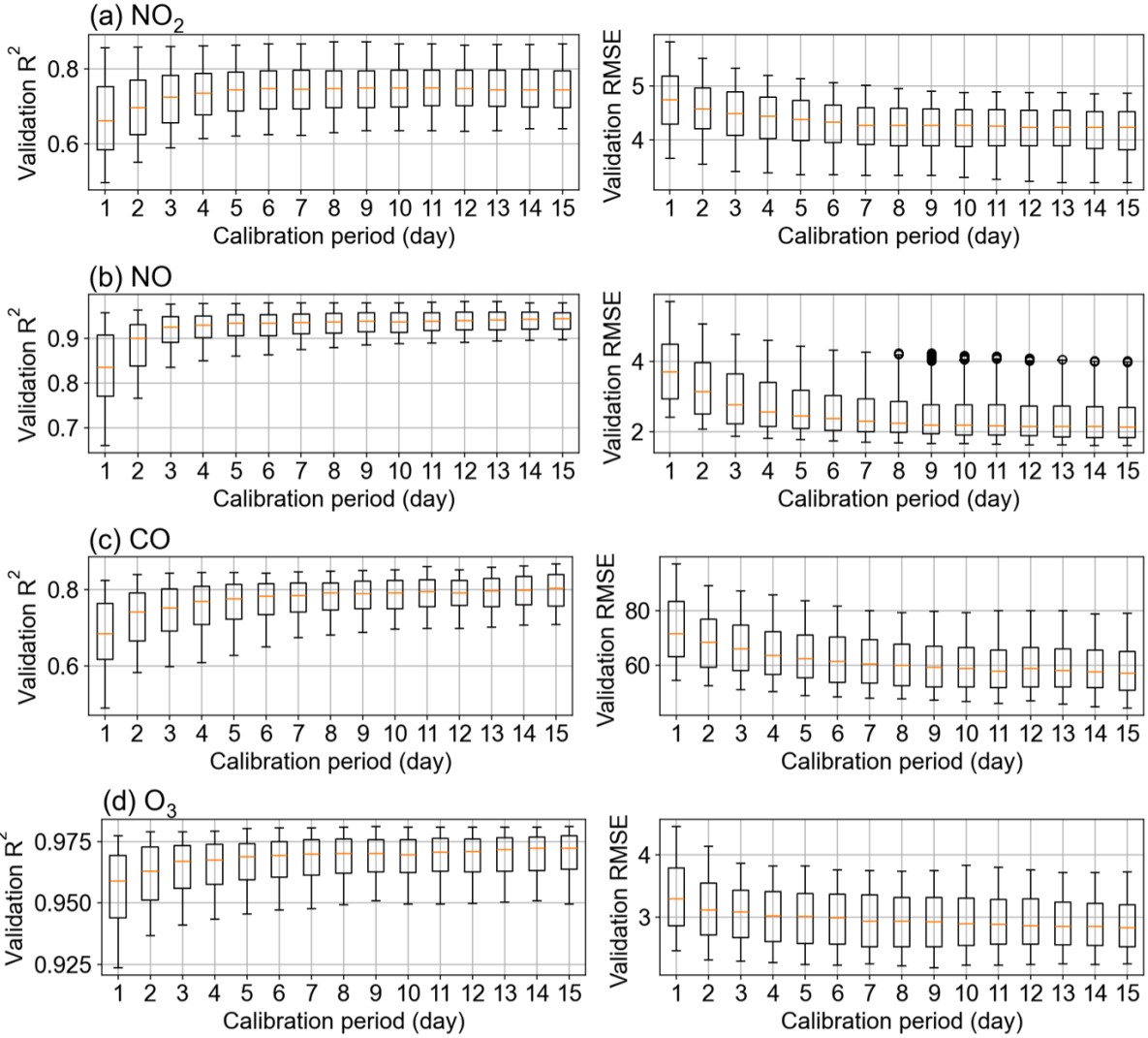

**Figure 4. The range of the validation $R^2$ and RMSE for a given calibration period for all MAS units consists of (a) NO$_2$, (b) NO, (c) CO, and (d) O$_3$ sensors. The vertical error bar is the 25%–75% distribution of $R^2$ and RMSE under different calibration periods.**

## 3.3 Impact of concentration range on sensor calibration

Another critical aspect is the impact of the concentration range experienced by the sensors during calibration periods. Figure S6 shows that MAS7 and MAS8 in the Shanghai campaign could achieve accurate and reliable calibration for NO and NO$_2$ within just a day, given their exposure to environments with significant concentration variability. Our second test examined the effect of the concentration range.

Samples in Figure 4 were grouped based on different concentration ranges, and the results were shown in Figure 5 and Figure S7 to explore the relationship between calibration period length, concentration range, and sensor validation performance, categorizing the MAS units accordingly. For $NO_2$ and NO sensors of MAS7 and MAS8, a separate analysis was essential due to their higher pollutant concentrations compared to other units, as detailed in Table 1. MASs 1-6 were evaluated together in Figure 5 under a lower concentration range, with 90% of $NO_2$ and NO ranges falling below 40 ppb and 50 ppb, respectively. MAS7 and MAS8 were assessed in Figure S7 under higher concentration ranges, where 90% of the readings for both gases exceeded these thresholds.

Figure 5 illustrates the calibration conditions at lower concentrations typical of environments like Hong Kong and Macau. The red zone of Figure 5, indicating higher $R^2$ values, is primarily concentrated in areas with wider concentration ranges. Specifically, when examining the performance of $NO_2$ sensors, the lowest $R^2$ value of 0.55 was recorded in the 0-10 ppb range, while the highest $R^2$ value of 0.75 was recorded in the >50 ppb range. When the calibration period is held constant, an increase in the concentration range boosts the validation $R^2$ from 0.55 to 0.75 with a notable turning point at 40 ppb. However, extending the calibration period without increasing the concentration range doesn't obviously improve the validation $R^2$. NO CO and $O_3$ also displayed patterns similar to $NO_2$, with $R^2$ improvements linked to wider concentration ranges. For all gases, the highest $R^2$ values were predominantly observed in the broadest concentration ranges. Therefore, achieving higher validation $R^2$ values above the median, such as $R^2 > 0.65$ for $NO_2$, $R^2 > 0.84$ for NO, $R^2 > 0.75$ for CO, and $R^2 > 0.95$ for $O_3$, requires significant concentration ranges, notably more than 40 ppb for $NO_2$ and 10 ppb for NO, 500 ppb for CO, and 20 ppb for $O_3$. Reaching these ranges allows the calibration coefficients to stabilize and align closely with those derived from year-long calibration results.

The recommended concentration ranges are 40 ppb, 10 ppb, 500 ppb, and 20 ppb for $NO_2$, NO, CO, and $O_3$, respectively. The differences in these concentration thresholds for various gas sensors may be attributed to the distribution characteristics of the gas pollutants in the surrounding environment. The NO concentration range of 10 ppb is the lowest, possibly due to the prevalence of high ambient NO concentrations frequently appearing in the form of peaks. When employing the 5th to 95th percentile as the criteria for concentration range, the NO range is observed to be the lowest among the gases. The higher concentration range analysis in Figure S7 shows that increasing the concentration range beyond 40 ppb for $NO_2$ and 50 ppb for NO does not improve validation $R^2$ values, further indicating a threshold in the concentration range beyond which no additional sensor performance benefits are observed. This underscores the inadequacy of merely extending the calibration duration, and it is crucial to ensure an adequate concentration range during the calibration period. But beyond a certain concentration range threshold, further increases in the calibration range do not lead to additional improvements in the calibration results.

It is important to acknowledge certain limitations in this section. The range of environmental concentrations tested was limited and may not encompass all possible calibration scenarios. Consequently, we lack sufficient data to support similar conclusions for environments with either significantly larger concentration ranges—such as those where NO, $NO_2$, and $O_3$ concentrations exceed 150 ppb—or those with consistently lower concentrations, where values remain below 10 ppb for extended periods.

While our findings are applicable to most similar or closely related concentration environments, further investigation is needed to validate these conclusions across a broader spectrum of calibration conditions.

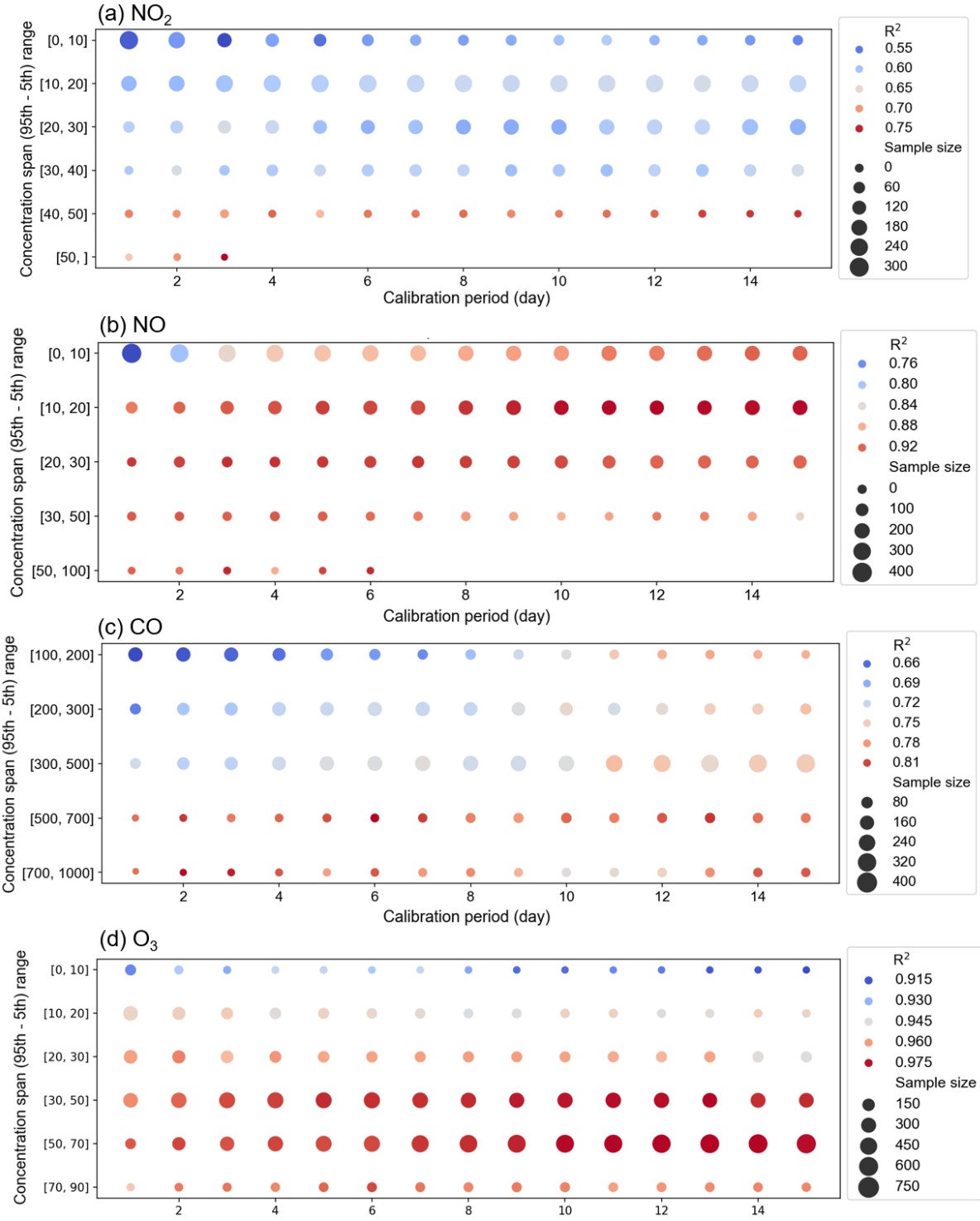

**Figure 5. (a) NO₂, (b) NO, (c) CO, and (d) O₃ bubble plot of median $R^2$ of MAS units 1–6 (as located in the low-concentration regions (Hong Kong and Macau)) and two factors: calibration period and concentration range. The size of the bubbles represents the number of samples. The color represents the median $R^2$ values in corresponding categories. Red represents the higher $R^2$ value, while blue represents the lower $R^2$ value.**


## 3.4 Impact of time averaging on sensor calibration

Another factor influencing calibration is the time averaging of the raw data, particularly for high-frequency measurements, taken at intervals of a minute or seconds. Performing temporal averaging is critical before formulating the calibration equation. As indicated in Table 1, only the reference data from Hong Kong was obtained at a one-minute temporal resolution. Thus,

only the data from MAS1 - 4 will be used for time averaging evaluation. The time averaging process aims to enhance the accuracy of calibration coefficients while ensuring a substantial data volume for a reliable calibration process.

Figure 6 presents results from two different perspectives: (a)-(c) focus on the time averaging analysis and the consistency of results across different sensors, while (d)-(f) emphasize the patterns observed under varying calibration periods. Figure 6(a)-(c) show the performance of the NO₂ sensors from MAS1 to MAS4 across different time intervals, ranging from one minute

to three hours. To eliminate the influence of the calibration period and adhere to the principles of single variable analysis, we utilized only 500 calibration samples from each MAS with a fixed calibration period of one day. The sensor and reference data for each calibration sample underwent time averaging across intervals of 1/3/5/7/9/11/30/60/120/180 minute(s). Subsequent calibration and validation led to the determination of the calibration slope, $R^2$ of the validation set, and RMSE for these time-averaged intervals. The results reveal a clear trend of improvement across all three metrics with increasing time averaging

intervals, particularly notable between the 1-minute and 5-minute intervals. All four MAS NO₂ sensors exhibit a consistent trend in this regard.

These findings are based on a calibration period of 1 day, and we extended the analysis to other calibration periods. Using MAS1 as an illustrative case, Figure 6(d)-(f) display the trends across different time averaging under various calibration periods. We derived the median values under each category. Analysis of Figure 6(e)'s vertical axis reveals that, for a one-day calibration

period, $R^2$ values improved post hourly ($R^2 = 0.68$) and 5-minute averaging ($R^2 = 0.66$) compared to the baseline 1-minute data ($R^2 = 0.59$), with a corresponding reduction in RMSE. For periods exceeding a day, median $R^2$ values exhibited a modest rise from 0.64-0.66 for 1-minute data to 0.68-0.70 for hourly data, suggesting the shorter the calibration period, the more pronounced the benefit of longer time averaging. Hence, calibrating with minute-level data over short periods of 1-3 days may lead to suboptimal validation performance. Similar trends were observed for NO and CO, as shown in Figures S8-S9; however,

the trend for O₃ shown in Figure S10 was less pronounced, with only the calibration slope exhibiting a similar pattern. This may be attributed to the unique characteristics of O₃ calculations (Eq. 2), where the influence of cross-interference from NO₂ affects the results, thereby masking the impact of time averaging.

The results indicate that data averaging over an hour are more suitable for calibration than minute-level data. As depicted in Figure 6 and Figures S8-S10, a critical juncture is identified at the 5-minute mark (highlighted by a green line with a star). After this point, the improvements in validation $R^2$ and RMSE become substantially less obvious. Thus, for data originally recorded at 1-minute intervals, applying a time averaging of 5 minutes or longer boosts the performance of the validation set, aligning the calibration coefficient more closely with the optimal one. The enhanced performance of hourly over minute-level data across various calibration periods warrants further investigation in the next section to understand the underlying factors.

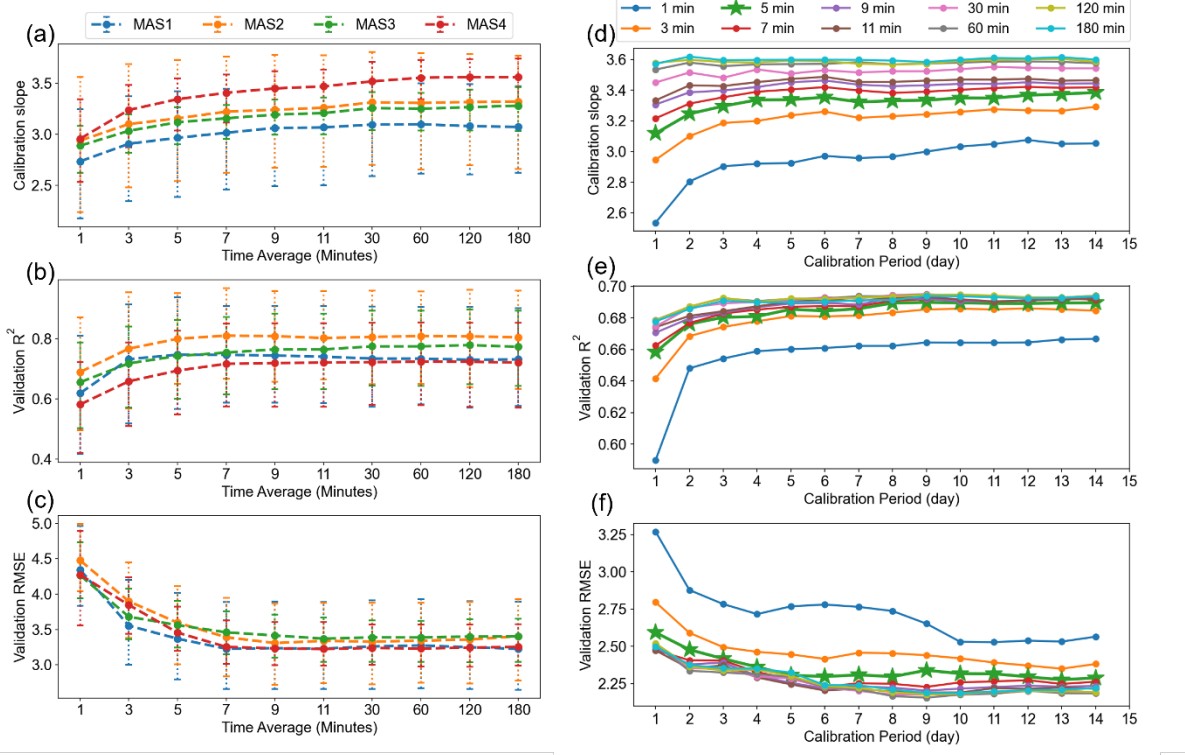

**Figure 6. (a)-(c) The potential range of calibration slope, the $R^2$, and the RMSE of the validation set for MASs 1-4 NO₂ sensors, under various time averaging with a calibration period of 1 day. Different colored lines represent the results of different MAS units. The vertical error bar is the 25%–75% distribution of the results under different categories. (e)-(f) The calibration slope median, the $R^2$ median, and the RMSE median of the validation set for MAS1 NO₂ sensors across all calibration periods, with different colors denoting time averages ranging from one minute to three hours.**

## 3.5 Potential causes of sensor calibration coefficient variation

We selected a sample from the MAS1 NO₂ sensor with a one-day calibration period to analyze the benefits of hourly over minute-level data averaging. Regression analysis between sensor and reference data was performed for both 1-minute and 1-

hour averages. Initially, data fitting during the calibration period was assessed. The time series plot in Figure 7(a) shows that both minute and hourly averaged data closely align with the reference. However, obvious differences emerged when computing the calibration equations separately for each time frame. The calibration slope for minute-level data (a = 2.70) was substantially lower than that for hourly data (a = 3.91), corroborating the trends noted in Figure 6. This discrepancy is evident in Figure 7(b), where the regression curves for minute-level and hourly data diverge. The orange line for hourly data intersects more

closely with the dense cluster of orange dots representing minute-level data, unlike the minute-level data's blue fitting line, which misses this dense area. In the validation phase, applying the distinct calibration coefficients derived for minute and hour averages to the next month's dataset also highlighted clear differences. Figure 7(c) and (d) illustrate that minute-level calibration coefficients (blue line) resulted in less consistent sensor data with the reference ($R^2 = 0.72$, RMSE = 5.60) than the hourly data ($R^2 = 0.93$, RMSE = 2.74), especially at lower concentrations.

The discrepancy between the two sets of calibration coefficients is further illustrated in the data distribution plots in Figure S11, where sensor and reference data distributions for varying time averaging lengths are compared. As the time averaging interval increases, the sensor data distribution more closely mirrors the reference data distribution. This observation supports the notion that time averaging can refine the accuracy of the calibration by aligning data distributions, leading to more precise calibration outcomes. This pattern consistently appeared across various samples, MAS units, and gases, as described in section

3.4, demonstrating the superior calibration accuracy achieved with longer averaging periods.

    Furthermore, we investigated the potential factors for the observed pattern by analyzing the residual term in sensor calibration model from the mathematical perspective. The detailed analysis is provided in Text S1 of the Supplementary Material. One plausible explanation is that the predictive capability of the calibration model using minute-level data may be compromised due to data noise. This noise can introduce variability that obscures underlying trends, ultimately leading to less reliable

regression results and hindering the model's ability to accurately capture the relationship between sensor and reference data. While this explanation is plausible, we currently lack specific insights into which influential factors may be affecting the regression model. This remains an area for further investigation in our future work.

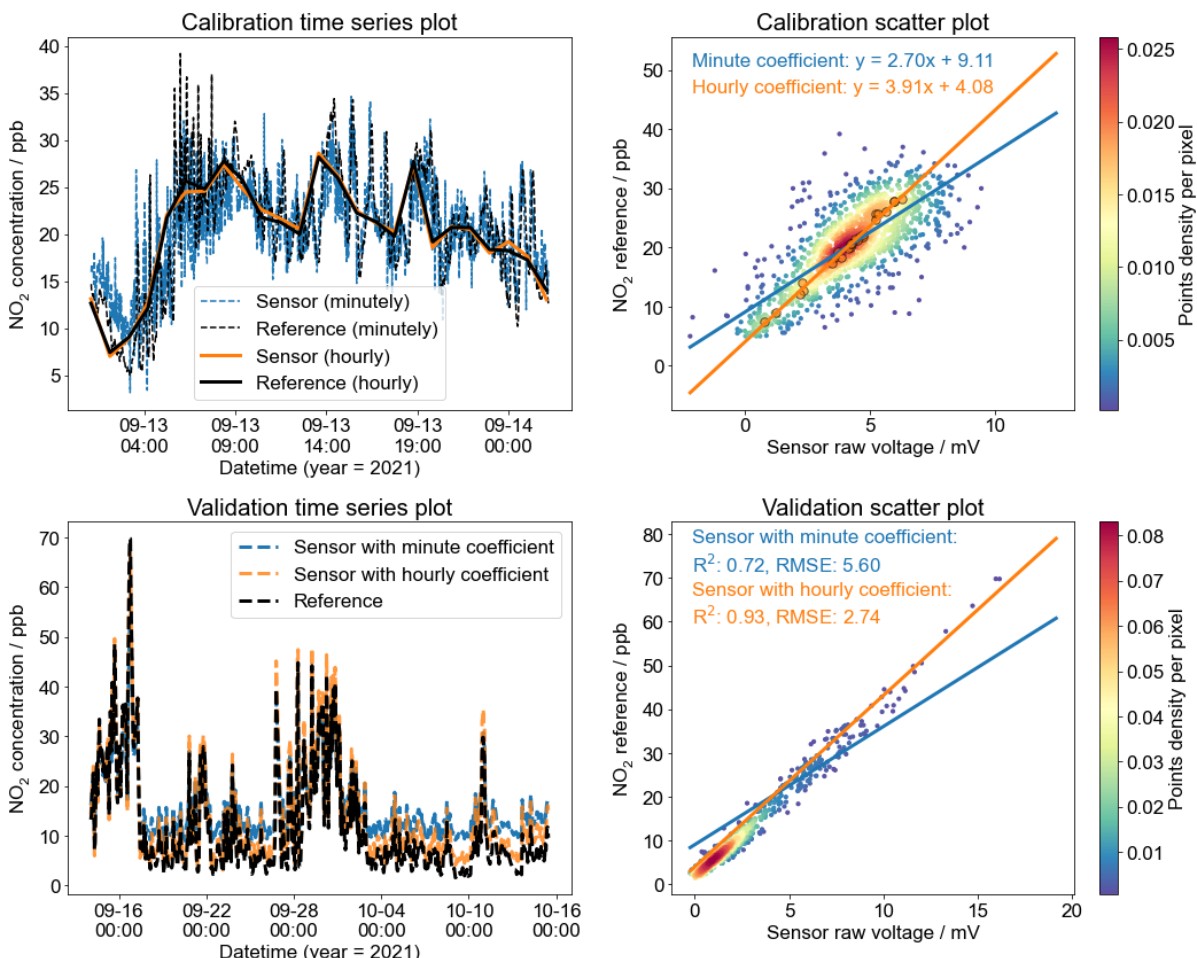

**Figure 7. One of the calibration samples for MAS1's NO₂ sensor with a calibration period of 1 day. (a) is the time series and (b) is the scatter plot of minute-level and hourly data for the NO₂ sensor and reference during the calibration period. (c) is the time series and (d) is the scatter plot of minute-level and hourly data for the NO₂ sensor and reference during the validation period. The color bars in (b) and (d) represent the sample size in each region.**

## 435 4. Conclusions

This study aimed to assess the performance of a novel dynamic baseline tracking method equipped with patented PDF gas sensors under different climate conditions, and to critically analyze three factors influencing the sensor calibration performance of PDF enabled NO₂, NO, CO, and O₃ sensors: calibration period, concentration range, and time averaging. By co-locating eight MAS units with reference analyzers in three cities over a period of up to 22 months, a comprehensive framework for

sensor calibration was established. The study utilized a dynamic baseline tracking method, enhancing the consistency between

MAS sensor data and reference measurements. This method effectively countered the impacts of temperature and RH, focusing on pollutant concentration as the primary factor for MAS performance assessment.

In the calibration period analysis, equations were derived from 500 randomly selected samples for each period ranging from 1 to 15 days, with subsequent evaluation against the validation data. Initial improvements in validation performance were notable within the first 1 to 3 days of the calibration period, stabilizing around 5 to 7 days. This pattern suggests that extending the calibration period beyond 7 days yields negligible benefits, hence a 5–7 days calibration period is advocated to reduce calibration coefficient errors.

The concentration range assessment indicated that broader ranges enhance the validation $R^2$ values across all gas sensors. This finding emphasizes the necessity of establishing a concentration range threshold to facilitate effective calibration. Optimal ranges were determined as over 40 ppb for $NO_2$, 10 ppb for NO, 500 ppb for CO, and 20 ppb for $O_3$, with these thresholds ensuring reliable calibration coefficients and minimizing uncertainty in the results.

Time averaging's impact on calibration was significant, with improved coefficients and validation performance as averaging intervals increased. The one-day calibration period showed the most substantial improvement, with hourly and 5-minute averages providing higher $R^2$ values than one-minute intervals. A 5-minute threshold emerged as critical, advocating for a minimum of 5-minute averaging to enhance calibration accuracy and align coefficients with the optimal standard.

This study offers comprehensive insights into calibrating electrochemical gas sensors, highlighting the calibration period, concentration range, and time averaging's roles. Recommended practices for optimal calibration include: (1) a calibration period of 5–7 days using hourly data, (2) a concentration variation range (5th to 95th percentile range) exceeding 40 ppb for $NO_2$, 10 ppb for NO, 500 ppb for CO, and 20 ppb for $O_3$, and (3) a time averaging of 5 minutes or longer, preferably utilizing hourly data. The findings highlight the importance of balancing these factors to achieve optimal calibration outcomes, while extending certain calibration aspects beyond recommended thresholds may not yield additional benefits.

Acknowledging the limitations of this study, which focused exclusively on our MAS sensor technology with its active flow gas sampler, it should be noted that the specific calibration protocol described may not be directly applicable to studies involving different sensor types, commercial sensor packages from various manufacturers, or different air sampling methods using passive samplers. Optimal calibration conditions may vary depending on the sensor's specific features and the calibration methods employed. For instance, regarding the optimized calibration period, a duration of at least 5 to 7 days is necessary for conditions with a narrower concentration range. In contrast, in locations with more polluted ambient environments, a shorter calibration duration of 1 to 3 days may be sufficient for effective sensor concentration calibration.

Future research endeavors should aim to diversify sensor types and increase the number of test sensors, thereby enhancing the generalizability and practicality of the findings. Nonetheless, the primary objective of this study is to provide methodological insights that can serve as a valuable reference for calibrating various sensor types. The developed dynamic baseline tracking method, along with the determined optimal calibration period, concentration range thresholds, and time averaging period, can inform and guide future research and calibration efforts for a wide range of sensors used in air quality monitoring. By

establishing a foundation for standardized calibration approaches, this study contributes to advancing sensor technologies and promoting the generation of reliable and comparable air quality data across diverse monitoring networks.

**CRediT authorship contribution statement**

**Han Mei:** Writing – original draft, Visualization, Methodology, Data curation, Conceptualization. **Peng Wei:** Writing – review & editing, Validation, Methodology, Conceptualization. **Meisam Ahmadi Ghadikolaei:** Writing – review & editing, Investigation. **Nirmal Kumar Gali:** Writing – review & editing, Visualization, Investigation. **Ya Wang:** Software, Methodology. **Zhi Ning:** Writing – review & editing, Validation, Supervision, Methodology, Conceptualization.

**Declaration of competing interest**

The authors declare that they have no known competing financial interests or personal relationships that could have appeared to influence the work reported in this paper.

**Code/Data availability**

Code and data will be made available on request.

**Acknowledgements**

The authors acknowledge the financial support received from the Research Grants Council of Hong Kong through the General Research Fund (16212022) and also acknowledge the support received from the Environmental Protection Department, HKSAR. The authors also appreciate the technical support offered by Sapiens Environmental Technology Co Limited, the manufacturer of the sensor devices used in the study.

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
