# Peer review of "Performance Validation and Calibration Conditions for Novel Dynamic Baseline Tracking Air Sensors in Long-term Field Monitoring"

_Atmospheric Measurement Techniques, 2024_

## Author Comment (AC1)

**Responses to reviewer #1 comments**

**Overview:**

*This work demonstrates an approach to calibration of a multi-gas sensor (NO, NO2, CO, O3) across three distinct locations, and by varying two driving factors (averaging time and concentration range) in improving calibration coefficients. There is also some discussion on a technique that purports to improve sensitivity and precision by a process that apparently removes water vapor, which is a known interfering compound for many of these electrochemical approaches. This work was performed at three different locations which were described by authors as being significantly different in climatology and in their pollution mixtures.*

**[Response]:** The authors really appreciate the reviewer for carefully reading our study and for their constructive comments. We believe that the improvements prompted by the thoughtful comments of reviewers have greatly strengthened the paper, for which we are grateful. Below we provide a point-by-point response (in blue text) to the reviewer's comments (black text) and summarize the changes that have been made in the revised track-changed manuscript. The quoted texts from the revised manuscript with the change tracked are in purple text.

**General comments:**

**[Comment]: 1.** *The work is generally sound, but viewed as somewhat incremental. It appears there are two distinct methods combined in this manuscript: the use of a water-removing improvement for sensitivity, and a simple linear modelling approach to assess adequacy of calibrations. Aside from using the same instrument platform, it is unclear how these two are closely related.*

**[Response]:** Thank you for your careful review. The linear modelling approach for the assessment in the study is established based on the dynamic baseline tracking technology. Here is the detailed explanation. While electrochemical sensors are well known for their linear response to the target pollutant concentrations, their sensor baseline signal are well demonstrated to have non-linear response to the temperature, relative humidity (RH), which significantly lowers their performance in ambient applications. Instead of relying solely on mathematical algorithms for sensor calibration, we assessed a novel dynamic baseline tracking technology designed to physically mitigate temperature and RH effects on sensor signals, allowing these kinds of gas sensor devices, i.e. Mini Air Stations (MAS, Sapiens), to output sensing data most directly related to the concentration signal. By isolating the non-linear influences of temperature and RH on sensor readings, this technology allowed us to focus exclusively on concentration calibration and enabled the development of a refined linear calibration model. Based on the linear calibration model, we further identify the critical factors that influence calibration quality, thus optimizing calibration conditions for $NO_2$, NO, CO, and $O_3$ electrochemical sensors.

We have added a detailed explanation in the revised manuscript as below:

**Line 63-81:**

This study focuses on electrochemical sensors, which are the most common type of air quality gas sensors. Laboratory studies of commercial electrochemical sensors have shown linear

correlations between current response and gas analyte concentration under stable temperature and relative humidity (RH) conditions (Mead et al. 2013; Collier-Oxandale et al. 2020; Wei et al. 2018; Zong et al. 2021). However, due to their electrochemical characteristics, these sensors often exhibit non-linear responses to variations in temperature and RH (Wei et al. 2018; Ariyaratne et al. 2023; Li et al. 2021), which can significantly impair their performance in real-world applications. In the past, most studies have adopted generic multiple linear regression (MLR) or machine learning models to calibrate raw sensor data, taking into account various complex variables such as temperature, RH, their gradient and cross-sensitivity to other pollutants (Datta et al., 2020; Han et al., 2021; Levy Zamora et al., 2023; Si et al., 2020; Topalovic et al., 2019; Wei et al., 2020; Zimmerman et al., 2018). These models, while comprehensive, often face limitations such as the risk of over-fitting, extensive training requirements, restricted applicability, and difficulties in replicating and scaling up for large sensor numbers. Furthermore, the complexity of machine learning models can pose significant barriers for everyday users.

Instead of relying solely on mathematical algorithms for sensor calibration, we investigated a dynamic baseline tracking technology designed to physically mitigate temperature and RH effects on sensor signals, allowing the sensor devices, Mini Air Stations (MASs), to observe data most directly related to the concentration signal. By isolating the non-linear influences of temperature and RH on sensor readings, this technology allowed us to focus exclusively on concentration calibration and enabled the development of a refined linear calibration model. Based on the linear calibration model, we further identify the critical factors that influence calibration quality, thus optimizing calibration conditions for $NO_2$, $NO$, $CO$, and $O_3$ electrochemical sensors.

**[Comment]: 2.** *There is a significant amount of this manuscript that discusses the employment of their 'dynamic baseline tracking method' to reduce water interference. However, aside from some clearly improved RMSE and R2 values when this approach was used, very little data are presented. Furthermore, this is approach simply improves sensitivity of an existing measurement method, which by definition, is incremental. This may indeed be a significant technological advancement, but as presented, the conclusions are not adequately supported by the data.*

**[Response]:** Thank you for pointing this out. We agree with the suggestion to provide more test data relevant to the methodology assessment and more explanation of the working principles for the readers to better understand the messages.

In the revised manuscript, we have added a new Figure 3 in Section 3.1 that illustrates the separate outputs from the PDF sensor, the ORG sensor, and the differential output between the paired ORG-PDF sensors. This addition, in conjunction with the conceptual diagram in Figure 2, aims to clarify how the PDF technology facilitates dynamic baseline tracking. We have also included more detailed analysis of the three separate outputs. From the PDF sensor output shown in Figure 3, we observed that the outputs for all gas pollutants did not exhibit a linear relationship with temperature or RH profiles. Different sensor types demonstrated distinct

response patterns to variations in temperature and RH. These findings highlight the complex non-linear characteristics of electrochemical sensors in relation to baseline dependence on these environmental factors. Then by subtracting the output of the PDF sensor from that of the ORG sensor, the resulting ORG – PDF output reveals a clear gas concentration profile that aligns closely with reference measurements. The significantly higher R² values and lower RMSE for the ORG-PDF sensor output compared to the ORG sensor output indicate that the influence of temperature and RH on sensor signals has been effectively eliminated. The above describes the performance validation of the dynamic baseline tracking method under field conditions, which is the primary application scenario of this study. Additionally, Section 3.1 includes data from laboratory tests conducted in environmental chambers (see Figure S3). It also provides an overview of the long-term (1-year) co-location performance data, as shown in Figures S4 and S5. The outcomes from both field and laboratory tests confirm that the dynamic baseline tracking method effectively neutralizes temperature and RH effects, primarily for $NO_2$, NO, and $O_3$ sensors, achieving desired performance while focusing primarily on concentration factors for subsequent analysis. We have rephrased the relevant statement to solve this comment. For details, please refer to section 3.1 in the revised manuscript.

To better convey the focus of this research and avoid any potential misunderstandings, we have revised the title of the manuscript to: "Performance Validation and Calibration Conditions for Novel Dynamic Baseline Tracking Air Sensors in Long-term Field Monitoring."

**[Comment]: 3.** *More interesting are the results of their statistical methods for calibrating across different time domains. The general conclusion, that collocations should be on the order of 5-7 days, is generally coherent with other findings, and is additional evidence that supports a broader convergence of calibration approaches in the lower cost sensor paradigm. The data are not entirely persuasive, in part because there was limited discussion (aside from the introduction) in how these approaches varied across diverse airsheds of Macau, Hong Kong, and Shanghai. Assessing the dynamic range of concentrations needed is certainly important, but so is assessing sensor averaging time performance in the presence (or absence) of co-pollutants that vary across space. It would seem reasonable that these are important interferents that may affect sensor performance. This work would be much stronger if separate analysis for different environmental conditions were presented rather than combined together – if the findings (of 5-7 day collocations) were robust across different airsheds, this would be a very important finding. I would assume that this is more nuanced, and one might find significant averaging time differences in locations with substantially different composition, just as the authors found with concentration loadings. But, unfortunately, one cannot gain this insight from the work as presented.*

**[Response]:** Thank you for your insightful and detailed comments. There are two reasons why limited discussion on how these approaches varied across diverse airsheds of Macau, Hong Kong, and Shanghai. One is the inclusion of Hong Kong, Macau, and Shanghai with their distinct climatic conditions primarily aims to validate the performance of MAS sensors employing the dynamic baseline tracking method under vastly varying temperature and RH conditions. Section 3.1 has provided test results and evidence showing effective mitigation of

the influence of temperature and RH, and the further analysis in following sections then focus on the impact of calibration conditions as the reviewer has pointed out. Secondly, due to the individual performance variability during the long-term deployment, we combined these sensors to analyze the impact of the three primary factors on calibration conditions, aiming to reduce the influence of other random errors by individual sensors.

The reviewer raises a valid point regarding the necessity of separate analyses of MAS sensor results across different regions. In response to this comment, while retaining the previous results, we have incorporated additional discussion on the results from each region within our analysis of the three impact factors. Firstly, regarding calibration period optimization, the previous conclusions about the 5-7 day collocations were based on an average pattern derived from the combined data of all sensors. We have now expanded our discussion to address the differences in calibration periods observed among various regions, as illustrated in Figure S6. The updated content is as follows:

**Line 296-307:**

The aforementioned results are based on an average pattern derived from the combined data of all sensors. Figure S6 presents the separate performance of all eight MAS sensors over varying calibration periods. The $NO_2$, NO, CO, and $O_3$ sensors in MAS1-4 in Hong Kong and MAS5-6 in Macau exhibited trends consistent with those shown in Figure 4. A noteworthy observation in Figure S6(a)-(b) is that the $NO_2$ and NO sensors in MAS7 and MAS8 of Shanghai campaign showed consistent performance over all calibration periods, lacking the trends observed in Figure 4. Considering that the NO and $NO_2$ concentrations in the site of Shanghai are significantly higher than those in the other two cities, it is hypothesized that the elevated pollutant concentrations at the Shanghai port provided a more favorable calibration condition, thereby diminishing the contribution of the calibration period. Thus, we conclude that for calibration condition with a narrower concentration range, a calibration period of at least 5 to 7 days is necessary, whereas more polluted ambient environments are more conducive to sensor concentration calibration. Despite the short calibration duration of 1–3 days, the extensive concentration range assessed contributed to more precise calibration coefficients and improved validation performance, as will be discussed in the next section.

Secondly, in the concentration range analysis, we discussed two distinct concentration scenarios: MASs 1-6 in Hong Kong and Macau were evaluated together in Figure 5 under a lower concentration range, with 90% of $NO_2$ and NO measurements falling below 40 ppb and 50 ppb, respectively. MAS7 and MAS8 deployed in Shanghai were assessed in Figure S7 under higher concentration ranges, where 90% of the readings for both gases exceeded these thresholds. The analysis of the lower concentration range reveals that the recommended concentration ranges are 40 ppb for $NO_2$, 10 ppb for NO, 500 ppb for CO, and 20 ppb for $O_3$. The higher concentration range analysis in Shanghai shows that increasing the concentration range beyond 40 ppb for $NO_2$ and 50 ppb for NO does not enhance validation $R^2$ values. The overarching finding emphasizes the importance of ensuring an adequate concentration range during the calibration period, but beyond a certain threshold, further increases in the calibration range do not yield additional improvements in calibration results. Given that it already includes several separate analyses, we have chosen not to add further discussion in section 3.3.

Finally, regarding the discussion on time averaging, only the reference data from Hong Kong was obtained at a one-minute temporal resolution, which limited our evaluation of time averaging to data of MAS1-4. Concerning your suggestion about "assessing sensor averaging time performance in the presence (or absence) of co-pollutants that vary across space," we are currently unable to draw definitive conclusions on this aspect. Our previous results utilized only one MAS as an example to demonstrate that applying a time averaging of 5 minutes or longer enhances sensor performance, bringing the calibration coefficients closer to optimal values. To address this, we have included results from additional sensors in Figure 6 as well as in Figures S8-S10, and we have expanded the discussion of results from different sensors in the main text, as detailed below:

**Line 363-386:**

As indicated in Table 1, only the reference data from Hong Kong was obtained at a one-minute temporal resolution. Thus, only the data from MAS1 - 4 will be used for time averaging evaluation. The time averaging process aims to enhance the accuracy of calibration coefficients while ensuring a substantial data volume for a reliable calibration process.

Figure 6 presents results from two different perspectives: (a)-(c) focus on the time averaging analysis and the consistency of results across different sensors, while (d)-(f) emphasize the patterns observed under varying calibration periods. Figure 6(a)-(c) show the performance of the $NO_2$ sensors from MAS1 to MAS4 across different time intervals, ranging from one minute to three hours. To eliminate the influence of the calibration period and adhere to the principles of single variable analysis, we utilized only 500 calibration samples from each MAS with a fixed calibration period of one day. The sensor and reference data for each calibration sample underwent time averaging across intervals of 1/3/5/7/9/11/30/60/120/180 minute(s). Subsequent calibration and validation led to the determination of the calibration slope, $R^2$ of the validation set, and RMSE for these time-averaged intervals. The results reveal a clear trend of improvement across all three metrics with increasing time averaging intervals, particularly notable between the 1-minute and 5-minute intervals. All four MAS $NO_2$ sensors exhibit a consistent trend in this regard.

These findings are based on a calibration period of 1 day, and we extended the analysis to other calibration periods. Using MAS1 as an illustrative case, Figure 6(d)-(f) display the trends across different time averaging under various calibration periods. We derived the median values under each category. Analysis of Figure 6(e)'s vertical axis reveals that, for a one-day calibration period, $R^2$ values improved post hourly ($R^2 = 0.68$) and 5-minute averaging ($R^2 = 0.66$) compared to the baseline 1-minute data ($R^2 = 0.59$), with a corresponding reduction in RMSE. For periods exceeding a day, median $R^2$ values exhibited a modest rise from 0.64-0.66 for 1-minute data to 0.68-0.70 for hourly data, suggesting the shorter the calibration period, the more pronounced the benefit of longer time averaging. Hence, calibrating with minute-level data over short periods of 1-3 days may lead to suboptimal validation performance. Similar trends were observed for NO and CO, as shown in Figures S8-S9; however, the trend for $O_3$ shown in Figure S10 was less pronounced, with only the calibration slope exhibiting a similar pattern. This may be attributed to the unique characteristics of $O_3$ calculations (Eq. 2), where

the influence of cross-interference from $NO_2$ affects the results, thereby masking the impact of time averaging.

[Figure]

**Figure 6. (a)-(c) The potential range of calibration slope, the $R^2$, and the RMSE of the validation set for MASs 1-4 $NO_2$ sensors, under various time averaging with a calibration period of 7 days. Different colored lines represent the results of different MAS units. The vertical error bar is the 25%–75% distribution of the results under different categories. (e)-(f) The calibration slope median, the $R^2$ median, and the RMSE median of the validation set for MAS1 $NO_2$ sensors across all calibration periods, with different colors denoting time averages ranging from one minute to three hours.**

**Special comments:**

**[Comment]: 1.** *L14, 77, 138. The authors routinely refer to this technology as 'patented' but it is not clear what the purpose of this statement is.*

**[Response]:** We apologize for any confusion. The manuscript removed redundant description of the methodology and only kept one such instance in Section 2.2 while first introducing the method.

**[Comment]: 2.** *L35-45: much of this is well established science, and could be reduced with appropriate referencing.*

**[Response]:** Thank you for your suggestion. We have streamlined the content to focus on essential information regarding calibration methods. We emphasized the limitations of laboratory-based approaches and highlighted the advantages of side-by-side calibration in real-world settings. This should enhance clarity and conciseness while retaining important references. We have made the following changes in accordance with your advice:

**Line 31-38:**

Common calibration methods include multi-point calibration with standard gases, controlled chamber calibration (Sousan et al. 2016; Papapostolou et al. 2017), on-site probe gas calibrations, and field side-by-side calibration (Bisignano et al. 2022; Holstius et al. 2014; Spinelle et al. 2015; 2017). The first three methods are laboratory-based methods or rely on standard gas, which inherently possess constraints and may not fully capture the intricate interactions of multiple pollutants and environmental factors encountered in situ. This limitation raises concerns about the applicability of calibration results obtained under controlled conditions to actual monitoring environments(Castell et al. 2017). An alternative approach is the side-by-side calibration, which involves the co-locating sensor systems with reference analyzers in real-world environmental settings for a designated duration.

**[Comment]: 3.** *L93-95: This is relatively well known methodology with Alphasense sensors; this could be refined and reduced.*

**[Response]:** Thank you for your suggestion. We have streamlined the text to reduce redundancy while retaining essential explanations for clarity as below.

**Line 92-95:**

This study focuses on electrochemical gas sensors for $NO_2$ (Alphasense $NO_2$-B43F), NO (Alphasense NO-B4), CO (Alphasense CO-B4), and $O_3$ (Alphasense OX-B431). Please note that the $O_3$ concentration is determined by calculating the difference between the readings of the oxidizing gas sensor (OX-B431) and the $NO_2$ sensor ($NO_2$-B43F).

**[Comment]: 4.** *L139: the authors refer to the approach as only allowing 'water molecules' to pass through a filter. Are all other gases excluded? This would be a very unusual method to assess water vapor interference.*

**[Response]:** Thank you for your review and we have given a more concise and clear explanation of the working principle in the revised manuscript. Briefly, the method is not essentially to remove water vapor but to use the PDF sensor to physically track the sensor's baseline signal due to varying environmental conditions including both temperature and humidity. The PDF sensor was tested against different gases and the results were presented in Section 3.1 demonstrating no target gases could induce the baseline signal from the PDF sensor. This added discussion would provide more evidence of the performance of the sensors in mitigating the environmental impact. We have added a detailed explanation in the revised manuscript as below:

**Line 141-146:**

This gas sensor system comprises a primary sensor – that is directly exposed to the air, capturing the original signal (designated as ORG) influenced by varying pollutants, temperature, and RH - and a proprietary pair differential filter sensor (designated as PDF) to track the dynamic baseline signal driven only by temperature and RH. The PDF sensor is equipped with a water molecule permeable membrane that allows the water vapor to penetrate through while filtering out the target gas modules from entering the sensor head. The

differential signal (measured in volts) between the ORG and PDF sensors decouples the temperature and humidity effects, yielding a pure signal that reflects target gas concentrations.

**[Comment]: 5.** *L171: The authors use a Python function to compute random numbers, but then note that this is to 'simulate real world sensor calibrations practices, and ensure randomness…' While this reviewer would certainly agree that random.choice() indeed chooses data randomly, it does not simulate anything.*

**[Response]:** Thank you for pointing this out. We acknowledge that the term "simulate" may have led to some ambiguity in our description. To clarify, our intention was to convey that by using the python function, we aim to replicate the variability and randomness of co-location timing that would be encountered in real-world sensor calibration practices. While the function itself generates random selections, the broader context of our study is to create hypothetical scenarios that reflect the diverse conditions under which calibration might occur in practice. We revised accordingly to better articulate this point as below, emphasizing that our approach is designed to mimic the randomness inherent in actual sensor calibration processes, rather than to simulate them in a strict sense.

**Line 180-183:**

We investigated calibration period scenarios ranging from 1 to 15 days. In each scenario, 500 samples were randomly selected using the numpy.random.choice() function in Python, ensuring randomness and independence in the selection of co-location timing. This approach is intended to create hypothetical scenarios that reflect the diverse conditions and variability under which calibration might occur in real-world sensor calibration practices.

**[Comment]: 6.** *L180: What does 'with superior validation performance..' mean?*

**[Response]:** To clarify, the phrase "with superior validation performance" refers to the situation where the calibration period results in higher $R^2$ values and lower RMSE values during the validation phase. This indicates that the calibrated sensor data aligns closely with the reference data, suggesting that the calibration period effectively captures the underlying relationship between the sensor and the reference measurements. We have revised accordingly to include this clarification:

**Line 190-193:**

Subsequently, these coefficients were validated using the following month's data by comparing the hourly calibrated sensor data and hourly reference data. A superior validation performance, indicated by higher $R^2$ values and lower RMSE values, suggests that the calibration period effectively captures the relationship between the calibrated sensor and the reference data, thereby indicating an optimal calibration duration.

**[Comment]: 7.** *L118: What do the authors mean by 'showcasing'? Reference monitoring sites are not normally a showcase, but focus on high quality empirical measurements. Consider revising this language.*

**[Response]:** Thank you for your feedback. We have revised the text as below to clarify that the reference monitoring site serves as a representative urban station, providing conditions suitable for sensor evaluation in a complex urban environment.

**Line 118-121:**

The first co-location campaign in Hong Kong involved the four MASs, each equipped with all four types of gas sensors ($NO_2$, $NO$, $CO$, and $O_3$), which were placed at the Tseung Kwan O AQMS (22.3716°E,114.1148°N) regulated by the Hong Kong Environmental Protection Department. This station serves as a representative urban site, providing conditions suitable for sensor evaluation in a complex urban environment.

**[Comment]: 8.** *Figure 2: The conceptual diagram does not add much to this paper, unless the focus of the paper were on method development for water interference signal removal.*

**[Response]:** We appreciate your feedback regarding Figure 2. We would like to clarify that the dynamic baseline tracking method is an important component of this study. Other reviewers also requested to add further details of the diagram. In our view, the conceptual diagram serves to illustrate how this method operates, enhancing the reader's understanding of its significance. In response to your comment, we have expanded the figure caption to provide more detailed descriptions of the content depicted in the diagram. Additionally, to emphasize the importance of this method, we have revised the title of the paper to "Performance Validation and Calibration Conditions for Novel Dynamic Baseline Tracking Air Sensors in Long-term Field Monitoring."

**Line 162-169:**

[Figure]

Figure 2. A conceptual diagram of the PDF-enabled MAS sensor device. In laboratory tests, standard gas with constant concentrations is periodically injected into the PDF and ORG sensors throughout varying temperature and RH cycles to investigate their effects on the sensor

performance. The PDF tracks the baseline signal driven only by temperature and RH, while the ORG sensor captures the concentration profile influenced by both the target gas module and environmental conditions. The differential signal between the ORG and PDF sensors decouples the baseline signal induced by temperature and RH, producing a pure signal that reflects the target gas concentrations. This concept is also applicable to ambient conditions, where the differential signal between the paired ORG and PDF sensors demonstrates the accuracy and robustness of PDF technology for ambient air monitoring.

**[Comment]: 9.** *Line 304: What do the authors mean by 'as determined in the just-obtained results'?*

**[Response]:** Thank you for your comment. We have revised the text to clarify that "just-obtained results" refers to the findings from the previous paragraph, specifically that achieving higher validation $R^2$ values requires significant concentration ranges: notably more than 40 ppb for $NO_2$, 10 ppb for NO, 500 ppb for CO, and 20 ppb for $O_3$. We indicate that the NO concentration range of 10 ppb is the lowest among these thresholds, and we discuss the reasons behind this observation.

**Line 337-340:**

The recommended concentration ranges are 40 ppb, 10 ppb, 500 ppb, and 20 ppb for $NO_2$, NO, CO, and $O_3$, respectively. The differences in these concentration thresholds for various gas sensors may be attributed to the distribution characteristics of the gas pollutants in the surrounding environment. Notably, the NO concentration range of 10 ppb is the lowest, possibly due to the prevalence of high ambient NO concentrations frequently appearing in the form of peaks.

**[Comment]: 10.** *Line 369: This sentence does not make sense. Isn't this always a plausible explanation for the failure of calibration models?*

**[Response]:** Thank you for pointing this out. In this section, we aim to explore the potential reasons for the variation in sensor calibration coefficients across different time averaging processes. In our case study, we observed that the regression performance of minute-level data is inferior to that of hourly-level data. Further analysis of the residuals from the mathematic perspective supports this observation. We suspect that data noise from minute-level data could be a contributing factor. Minute-level data can introduce variability that obscures underlying trends, potentially leading to less reliable regression results. While this explanation is plausible, we currently lack specific insights into which influential factors may be affecting the regression model. This remains an area for further investigation in our future work. Following this comment, we have revised accordingly to include this clarification:

**Line 420-426:**

Furthermore, we investigated the potential factors for the observed pattern by analyzing the residual term in sensor calibration model from the mathematical perspective. The detailed analysis is provided in Text S1 of the Supplementary Material. One plausible explanation is

that the predictive capability of the calibration model using minute-level data may be compromised due to data noise. This noise can introduce variability that obscures underlying trends, ultimately leading to less reliable regression results and hindering the model's ability to accurately capture the relationship between sensor and reference data. While this explanation is plausible, we currently lack specific insights into which influential factors may be affecting the regression model. This remains an area for further investigation in our future work.

**Editorial/Minor comments:**

**[Comment]: 1.** *L66: typo on 'more easily to be standardized', and needs clarification.*

**[Response]:** Thank you for your feedback regarding the phrasing "more easily to be standardized." We have revised accordingly for clarity. The updated text now highlights that the range of pollutant concentrations and the selection of time averaging length for raw data are more straightforward to standardize and quantify compared to other factors, as they can be defined with specific numerical values and consistent measurement protocols.

**Line 56-62:**

While these studies have offered valuable insights into sensor field calibration conditions, more discussion is needed on other calibration factors, particularly the range of pollutant concentrations during the calibration period and the selection of time averaging length for raw data before calibration. These two factors are more straightforward to standardize and quantify compared to other factors, as they can be defined with specific numerical values and consistent measurement protocols, making it easier to compare results across different studies and ensure reliable calibration outcomes.

**[Comment]: 2.** *While there are few specific editorial comments to address, the manuscript has a substantial amount of indirect language, including many unnecessary linguistic flourishes. The writing is far too verbose, and makes the work laborious to read. There are periods in which a number of sentences begin with unneeded adverbs (e.g. Line 20, 99, 307 all start with 'moreover; L 304, 397 begin with 'notably', L47, 283, 306 all begin with "consequently'). In fact, in the paragraph that begins at line 304, every sentence begins with this unneeded language (Additionally, Notably, Consequently, Moreover, Overall, However).*

**[Response]:** Thank you for detecting the language issues. We acknowledge that some sections contained verbose language and unnecessary adverbs, which could hinder readability. After a thorough review of the manuscript by native English speakers, we have removed most of these linking words to enhance clarity and reduce verbosity. We have retained essential linking words, such as 'however,' to emphasize transitions and maintain the text's flow.

---

## Author Comment (AC2)

**Responses to reviewer #2 comments**

**Overview:**

*This study is a thorough analysis of several factors influencing the calibration of low-cost sensors: concentration range, calibration duration, and time averaging, and provides recommendations for each. The study utilizes the MAS sensors, which have a built-in in 'dynamic baseline tracking' feature that promises to eliminate (reduce?) the effects of environmental conditions such as temperature and humidity.*

**[Response]:** We wish to thank the reviewer for the positive and valuable comments. We have carefully considered all suggestions when preparing this response and revising the manuscript. The incorporation of the reviewer's suggestions has led to a substantially improved manuscript, for which we are grateful. Below we provide a point-by-point response (in blue text) to the reviewer's comments (black text) and summarize the changes that have been made in the revised track-changed manuscript. The quoted texts from the revised manuscript with the change tracked are in purple text.

On the evidence of elimination or reduction of the environmental impact, we have given a detailed response in Comment 3. In general, we agree with the reviewer that 'elimination' is a strong claim and the current evidence from our long-term tests has only shown PDF-enabled sensors have very significant improvement in mitigating the environmental impact in the range of test conditions, while other more diverse and extreme conditions may be further investigated in the future to verify the performance of PDF sensors.

**General comments:**

**[Comment]: 1.** *The dynamic baseline tracking method described in section 2.2 is fascinating, and more detail in the explanation would help the reader understand it better. How does the PDF work, and can any information be shared on its accuracy at filtering out gases? Is it more or less accurate for any specific gases?*

**[Response]:** Thank you for your careful review. We have further clarified the details of the dynamic baseline tracking method as the reviewer has suggested. The method is made possible by the pair differential filter (PDF) technology from the manufacturer and we also gave more details in the responses to explain the working principle of PDF sensors. Our prior laboratory evaluations have demonstrated the effectiveness of the PDF sensor in filtering out target gases while maintaining the free passage of water molecules. For instance, when target gases such as CO, NO, $NO_2$, $O_3$, and $SO_2$ at varying concentrations are injected into the PDF sensors, no sensor response signal is observed showing the effectiveness of the method. The revised manuscript is as below:

**Line 135-145:**

The sensor device (MAS, Sapiens) has deployed a novel gas sensing technology that enables the isolation of the concentration signal from environmental variables of temperature and RH through a patented dynamic baseline tracking method by the manufacturer, which operates by differentiating the varying environment and target pollutant induced sensor signals using a dual-sensor module. Figure 2 shows the conceptual diagram of MAS sensor module and

general working principle of the dynamic baseline tracking method. This gas sensor system comprises a primary sensor – that is directly exposed to the air, capturing the original signal (designated as ORG) influenced by varying pollutants, temperature, and RH - and a proprietary pair differential filter sensor (designated as PDF) to track the dynamic baseline signal driven only by temperature and RH. The PDF sensor is equipped with water molecule permeable membrane that allows the water vapor to penetrate through while filtering out the target gas modules from entering the sensor head. The differential signal (measured in volts) between the ORG and PDF sensors decouples the temperature and humidity effects, yielding a pure signal that reflects target gas concentrations.

**[Comment]: 2.** *Figure 2 is helpful for understanding this, but in the upper right plot, is there really zero difference between ORG and PDF in the lab? If not, a similar figure showing real data from the lab is important for readers understand how perfect or imperfect the method is, even if in the supplemental. It is not clear from the figure what "laboratory conditions" in the right panel means – were temperature, pressure, or humidity held constant, or were all fluctuating?*

**[Response]:** We appreciate your feedback regarding Figure 2. In response to your comment, we have reorganized the figure content and expanded the figure caption to provide more detailed descriptions. As the figure serves as a conceptual diagram illustrating the method, it does not specify the laboratory conditions in detail. To clarify, the tests were conducted under varying temperature and RH cycles, while pressure was not considered in the tests. We have included the test data from laboratory tests in the supplementary materials, along with an explanation of the tests provided in Lines 253-259. The updated manuscript includes a detailed explanation as follows:

**Line 165-174:**

[Figure]

Figure 2. A conceptual diagram of the PDF-enabled MAS sensor device. In laboratory tests, standard gas with constant concentrations is periodically injected into the PDF and ORG sensors throughout varying temperature and RH cycles to investigate their effects on the sensor performance. The PDF tracks the baseline signal driven only by temperature and RH, while the

ORG sensor captures the concentration profile influenced by both the target gas module and environmental conditions. The differential signal between the ORG and PDF sensors decouples the baseline signal induced by temperature and RH, producing a pure signal that reflects the target gas concentrations. This concept is also applicable to ambient conditions, where the differential signal between the paired ORG and PDF sensors demonstrates the accuracy and robustness of PDF technology for ambient air monitoring.

**Line 253-259:**

Additionally, laboratory tests in environmental chambers assessed the MAS NO sensor (Figure S3), exposing it to broad temperature (0°C to 30°C) and RH (10% to 90%) ranges. Despite these fluctuations, MAS sensors maintained consistent and stable readings after applying the dynamic baseline tracking method, as shown in Figure S3(b), with concentration steps from 50 to 300 ppb. The outcomes from both field and laboratory tests confirm that the dynamic baseline tracking method effectively neutralizes temperature and RH effects, primarily for $NO_2$, NO, and $O_3$ sensors, achieving desired performance while focusing primarily on concentration factors for subsequent analysis. Similar pre-tests were also conducted with the MAS units in Macau and Shanghai to assess the effectiveness of the dynamic baseline tracking method.

[Figure]

Figure S3. (a) Laboratory environmental chamber setup and (b) the response of 3 MASs' NO sensors under multiple point concentrations in laboratory temperature and humidity test.

**[Comment]: 3.** *The authors later state that, "the influence of temperature and RH on sensor signals has been eliminated". Can you prove to the reader with real data that this is entirely eliminated, or to a certain extent eliminated? In theory, Figure 3 could help answer this for the ambient data, but it is hard to read. Figure 3 panel F is the only one I can somewhat make out the difference between solid and dotted lines for. Moving the black reference data to the back of these plots might help make the other colors and lines more visible, but additional edits might be necessary for readability.*

**[Response]:** Thank you for your valuable suggestion. To enhance the visibility of Figure 3, we have updated it in Section 3.1 to focus on one of the four tested MAS sensors as a representative example of PDF technology's robustness from our test results. The previous version has been moved to the supplementary material as a consistency overview across all four MAS performances. The new Figure 3 separately illustrates the outputs from the PDF sensor, the ORG sensor, and the differential output between the paired ORG-PDF sensors. This addition,

in conjunction with the conceptual diagram in Figure 2, aims to clarify how the PDF technology facilitates dynamic baseline tracking method.

Additionally, we have included more detailed analysis of the three separate outputs. From the PDF sensor output shown in Figure 3, we observed that the outputs for all gas pollutants did not exhibit a linear relationship with temperature or RH profiles. Different sensor types demonstrated distinct response patterns to variations in temperature and RH. These findings highlight the complex non-linear characteristics of the regular electrochemical sensors in relation to baseline dependence on these environmental factors, while the PDF enabled sensor output reveals a clear gas concentration profile that aligns closely with reference data. The significantly higher $R^2$ values and lower RMSE for the ORG-PDF sensor output compared to the ORG sensor output indicate that the influence of temperature and RH on sensor signals has been effectively mitigated. Additionally, Section 3.1 includes data from laboratory tests conducted in environmental chambers (see Figure S3). It also provides an overview of the long-term (1-year) co-location performance data, as shown in Figures S4 and S5. The outcomes from both field and laboratory tests confirm that the dynamic baseline tracking method effectively neutralizes temperature and RH effects, primarily for $NO_2$, NO, and $O_3$ sensors, achieving desired performance while focusing primarily on concentration factors for subsequent analysis. We agree with the reviewer that 'elimination' is a strong claim and the current evidence from our long terms tests have only shown PDF enabled sensors have very significant improvement in mitigating the environmental impact in the range of test conditions, while other more diverse and extreme conditions may be further investigated in the future to verify the performance of PDF sensors. We have revised the manuscript and use 'significantly mitigated' instead of 'eliminated' to be in alignment with the tests in the study.

To better convey the focus of this research and avoid any potential misunderstandings, we have revised also the title of the manuscript to: "Performance Validation and Calibration Conditions for Novel Dynamic Baseline Tracking Air Sensors in Long-term Field Monitoring." We revised the relevant statement in the revised manuscript as follows:

**Line 229-252:**

We tested four MAS units and presented findings from this one MAS as an example to evaluate the robustness of the PDF technology. During the 15-day pre-test in the summer (June 1-15, 2019), temperatures varied between 28 ˚C and 42 ˚C, with RH levels from 45% to 87%. The outputs from the PDF sensor, the ORG sensor, and the differential output between the paired ORG - PDF sensor are illustrated separately in Figure 3(a)-(d). The voltage signals from the PDF and ORG sensors were converted into concentration outputs using coefficients derived from Eq. (1). As shown in the figure, even during the typical ambient concentration ranges, the accuracy of the ORG sensor outputs for gases other than CO was notably poor, primarily due to significant influences from field temperature and RH. It was observed that the PDF sensor outputs for all gas pollutants did not exhibit a linear relationship with temperature or RH profiles. Different sensor types demonstrated distinct response patterns to variations in temperature and RH, highlighting the complex non-linear characteristics of electrochemical sensors in relation to baseline dependence on these environmental factors.

With the PDF enabled sensors, the physical separation of the climatic driven baseline and target gas driven sensitivity is demonstrated to be feasible and effective. By subtracting the output of the PDF sensor from that of the ORG sensor, the resulting ORG – PDF output reveals a clear gas concentration profile that aligns closely with reference measurements. This relationship is illustrated in the scatter plots presented in Figure 3(f)-(i). For $NO_2$, the ORG – PDF sensors showed stronger performance, with a high $R^2$ (0.99) and low RMSE (0.94), compared to the lower $R^2$ (0.44) and higher RMSE (5.80) for the ORG sensors without the PDF module. For NO and $O_3$, the ORG – PDF sensors also demonstrated stronger performance compared to the ORG sensors without the PDF module. Specifically, the ORG – PDF sensors had strong $R^2$ (0.97 for both NO and $O_3$) and low RMSEs (1.72 for NO, 1.05 for $O_3$), while the ORG sensors without the PDF module had weaker $R^2$ (0.73 for NO, 0.59 for $O_3$) and higher RMSEs (5.37 for NO, 4.18 for $O_3$). For CO, the sensors exhibited comparable performance, with $R^2$ around 0.93-0.94 and RMSE values between 16.70-19.00, regardless of the PDF module. We tested four MASs and the other PDF enabled sensors were shown in Figure S2. Their data quality performance has been consistent with the findings reported data here. These significant discrepancies between the ORG sensor output and ORG – PDF sensor output, especially for NO, $NO_2$, and $O_3$, highlight the importance of the dynamic baseline tracking method in improving the accuracy and reliability of measurements, notably under low concentration conditions influenced by temperature and RH.

[Figure]

Figure 3. (a-d) Performance validation of the MAS's ORG and PDF sensors for detecting $NO_2$, NO, CO, and $O_3$ under field conditions in 2019. (e) Displays the temperature and RH measured inside the MAS gas sensor modules. (f-i) Compares the readings from the ORG sensor and the MAS PDF-enabled sensor with reference measurements.

For details, please refer to section 3.1 in the revised manuscript.

**[Comment]: 4.** *The analyses of concentration range, calibration period, and time averaging are thoughtful and well-explained. The R2 and RMSE of the validation data are reported. Would these results change if you applied the best model from one location and applied it to another location? As the authors state, it is important to replicate the conditions of the deployment to the best extent possible during the colocation, but lack of availability of reference instruments in certain locations can make this difficult. The analysis shown here could be made more practical by showing examples of how these trends may deviate as low-cost sensor users frequently have to adhere to non-ideal constraints. The recommendations provided for each of these are well thought out for the best case scenario of being able to co-locate exactly where the deployment will take place, but I am left wondering if these recommendations would still apply in a real-world scenario.*

**[Response]:** Thank you for your insightful and detailed comments. The reviewer raises a valid point regarding whether these results vary across diverse locations. The strategically deployed eight sensor devices in three different locations allow us to verify whether optimized calibration conditions differ under varying environmental settings. Previously, we combined these sensors together for analysis, and there was limited discussion on how these results varied across diverse sensors. In response to this comment, while retaining the previous results, we have incorporated separate analyses of each MAS sensor results within our analysis of the three impact factors.

Firstly, regarding calibration period optimization, the previous conclusions about the 5-7 day collocations were based on an average pattern derived from the combined data of all sensors. We have now expanded our discussion to address the differences in calibration periods observed among various regions, as illustrated in Figure S6. The updated content is as follows:

**Line 296-307:**

The aforementioned results are based on an average pattern derived from the combined data of all sensors. Figure S6 presents the separate performance of all eight MAS sensors over varying calibration periods. The $NO_2$, NO, CO, and $O_3$ sensors in MAS1-4 in Hong Kong and MAS5-6 in Macau exhibited trends consistent with those shown in Figure 4. A noteworthy observation in Figure S6(a)-(b) is that the $NO_2$ and NO sensors in MAS7 and MAS8 of Shanghai campaign showed consistent performance over all calibration periods, lacking the trends observed in Figure 4. Considering that the NO and $NO_2$ concentrations in the site of Shanghai are significantly higher than those in the other two cities, it is hypothesized that the elevated pollutant concentrations at the Shanghai port provided a more favorable calibration condition, thereby diminishing the contribution of the calibration period. Thus, we conclude that for calibration condition with a narrower concentration range, a calibration period of at least 5 to 7 days is necessary, whereas more polluted ambient environments are more conducive to sensor concentration calibration. Despite the short calibration duration of 1–3 days, the extensive concentration range assessed contributed to more precise calibration coefficients and improved validation performance, as will be discussed in the next section.

Secondly, in the concentration range analysis, we discussed two distinct concentration scenarios: MASs 1-6 in Hong Kong and Macau were evaluated together in Figure 5 under a

lower concentration range, with 90% of $NO_2$ and NO measurements falling below 40 ppb and 50 ppb, respectively. MAS7 and MAS8 deployed in Shanghai were assessed in Figure S7 under higher concentration ranges, where 90% of the readings for both gases exceeded these thresholds. The analysis of the lower concentration range reveals that the recommended concentration ranges are 40 ppb for $NO_2$, 10 ppb for NO, 500 ppb for CO, and 20 ppb for $O_3$. The higher concentration range analysis in Shanghai shows that increasing the concentration range beyond 40 ppb for $NO_2$ and 50 ppb for NO does not enhance validation $R^2$ values. The overarching finding emphasizes the importance of ensuring an adequate concentration range during the calibration period, but beyond a certain threshold, further increases in the calibration range do not yield additional improvements in calibration results. Given that it already includes several separate analyses, we have chosen not to add further discussion in section 3.3.

Finally, regarding the discussion on time averaging, only the reference data from Hong Kong was obtained at a one-minute temporal resolution, which limited our evaluation of time averaging to data of MAS1-4. Our previous results utilized only one MAS as an example to demonstrate that applying a time averaging of 5 minutes or longer enhances sensor performance, bringing the calibration coefficients closer to optimal values. To address this, we have included results from additional sensors in Figure 6 as well as in Figures S8-S10, and we have expanded the discussion of results from different sensors in the main text, as detailed below:

**Line 363-386:**

As indicated in Table 1, only the reference data from Hong Kong was obtained at a one-minute temporal resolution. Thus, only the data from MAS1 - 4 will be used for time averaging evaluation. The time averaging process aims to enhance the accuracy of calibration coefficients while ensuring a substantial data volume for a reliable calibration process.

Figure 6 presents results from two different perspectives: (a)-(c) focus on the time averaging analysis and the consistency of results across different sensors, while (d)-(f) emphasize the patterns observed under varying calibration periods. Figure 6(a)-(c) show the performance of the $NO_2$ sensors from MAS1 to MAS4 across different time intervals, ranging from one minute to three hours. To eliminate the influence of the calibration period and adhere to the principles of single variable analysis, we utilized only 500 calibration samples from each MAS with a fixed calibration period of one day. The sensor and reference data for each calibration sample underwent time averaging across intervals of 1/3/5/7/9/11/30/60/120/180 minute(s). Subsequent calibration and validation led to the determination of the calibration slope, $R^2$ of the validation set, and RMSE for these time-averaged intervals. The results reveal a clear trend of improvement across all three metrics with increasing time averaging intervals, particularly notable between the 1-minute and 5-minute intervals. All four MAS $NO_2$ sensors exhibit a consistent trend in this regard.

These findings are based on a calibration period of 1 day, and we extended the analysis to other calibration periods. Using MAS1 as an illustrative case, Figure 6(d)-(f) display the trends across different time averaging under various calibration periods. We derived the median values under each category. Analysis of Figure 6(e)'s vertical axis reveals that, for a one-day

calibration period, $R^2$ values improved post hourly ($R^2$ = 0.68) and 5-minute averaging ($R^2$ = 0.66) compared to the baseline 1-minute data ($R^2$ = 0.59), with a corresponding reduction in RMSE. For periods exceeding a day, median $R^2$ values exhibited a modest rise from 0.64-0.66 for 1-minute data to 0.68-0.70 for hourly data, suggesting the shorter the calibration period, the more pronounced the benefit of longer time averaging. Hence, calibrating with minute-level data over short periods of 1-3 days may lead to suboptimal validation performance. Similar trends were observed for NO and CO, as shown in Figures S8-S9; however, the trend for $O_3$ shown in Figure S10 was less pronounced, with only the calibration slope exhibiting a similar pattern. This may be attributed to the unique characteristics of $O_3$ calculations (Eq. 2), where the influence of cross-interference from $NO_2$ affects the results, thereby masking the impact of time averaging.

[Figure]

**Figure 6. (a)-(c) The potential range of calibration slope, the $R^2$, and the RMSE of the validation set for MASs 1-4 $NO_2$ sensors, under various time averaging with a calibration period of 1 day. Different colored lines represent the results of different MAS units. The vertical error bar is the 25%–75% distribution of the results under different categories. (e)-(f) The calibration slope median, the $R^2$ median, and the RMSE median of the validation set for MAS1 $NO_2$ sensors across all calibration periods, with different colors denoting time averages ranging from one minute to three hours.**

[**Comment**]: 5. *The limitations and future works could be expanded into their own section instead of lumped into the conclusion, since these topics have not been discussed earlier in the paper. Acknowledgement of the practicalities of variation in sensor co-location vs deployment locations could also be expanded upon here.*

[**Response**]: We appreciate your suggestion regarding the limitations part. We would like to clarify that the limitations we discussed here are intended to highlight important considerations for the application of our findings in various scenarios. Given the diversity of sensor types,

commercial sensor packages from different manufacturers, and even varying air sampling methods, the corresponding calibration protocols can differ significantly. We believe it is crucial to clearly state that optimal calibration conditions may vary depending on the specific features of the sensor and the calibration methods employed. The primary objective of this study is to provide methodological insights that can serve as a valuable reference for calibrating various sensor types. The developed dynamic baseline tracking method, along with the established optimal calibration period, concentration range thresholds, and time averaging period, can inform and guide future research and calibration efforts for a wide range of sensors used in air quality monitoring.

In light of this comment, we have incorporated an example into the limitations discussion and also expanded some limitation statement in the results section, as detailed below:

**Line 464-467:**

Optimal calibration conditions may vary depending on the sensor's specific features and the calibration methods employed. For instance, regarding the optimized calibration period, a duration of at least 5 to 7 days is necessary for conditions with a narrower concentration range. In contrast, in locations with more polluted ambient environments, a shorter calibration duration of 1 to 3 days may be sufficient for effective sensor concentration calibration.

**Line 348-353:**

It is important to acknowledge certain limitations in this section. The range of environmental concentrations tested was limited and may not encompass all possible calibration scenarios. Consequently, we lack sufficient data to support similar conclusions for environments with either significantly larger concentration ranges—such as those where $NO$, $NO_2$, and $O_3$ concentrations exceed 150 ppb—or those with consistently lower concentrations, where values remain below 10 ppb for extended periods. While our findings are applicable to most similar or closely related concentration environments, further investigation is needed to validate these conclusions across a broader spectrum of calibration conditions.